# Exploring the space of self-reproducing ribozymes using generative models

Camille N. Lambert [1,8], Vaitea Opuu [1,2,8], Francesco Calvanese[1,3,8], Polina Pavlinova [1], Francesco Zamponi [4], Eric J. Hayden [5,6], Martin Weigt[3], Matteo Smerlak [1,2,7] & Philippe Nghe [1] ✉

Estimating the plausibility of RNA self-reproduction is central to origin-of-life scenarios. However, this property has been shown in only a handful of catalytic RNAs. Here, we compare models for their generative power in diversifying a reference ribozyme, based on statistical covariation and secondary structure prediction, and experimentally test model predictions using high-throughput sequencing. Leveraging statistical physics methods, we compute the number of ribozymes capable of autocatalytic self-reproduction from oligonucleotide fragments to be over $10^{39}$, with sequences found up to 65 mutations from the original sequence and 99 mutations away from each other, far beyond the 10 mutations achieved by deep mutational scanning. The findings demonstrate an efficient method for exploring RNA sequence space, and provide quantitative data on self-reproducing RNA that further illuminates the potential pathways to abiogenesis.

The RNA world hypothesis surmises that early evolving systems consisted of reactions between RNAs catalyzed by RNAs[1]. In this scenario, the first ribozymes (catalytic RNAs) would have arisen by chance from random polymerization, possibly helped by covalent assembly reactions between partially complementary RNAs[2,3]. Among those, certain ribozymes could catalyze their own production, a process called autocatalysis[4,5], which has been tested experimentally. For instance, ligase ribozymes selected in vitro were shown to catalyze the joining of two RNA oligonucleotides, such that the product forms an additional copy of the ribozyme[6,7]. In another system, naturally occurring self-splicing group I intron RNAs were engineered to make copies of themselves by splicing together four oligonucleotide substrates[8]. It is envisioned that a gradual process of evolution could have led from autocatalytic RNAs[9] to template-based replication of RNA catalyzed by polymerase ribozymes[10,11], thereby initiating a mode of evolution similar to biological evolution. While some ribozymes relevant to the origin of life still need to be discovered, another important question is whether there exists a diversity of sequences that can carry out a given

activity among all possible sequences (the sequence space), an ensemble called the *neutral network*, or *neutral set*[12]. In particular, the larger the diversity of self-reproducing RNAs, the more likely are transitions between self-reproducing systems, thus enabling primordial modes of evolution[13].

One challenge to studying the neutral set of ribozymes is the extremely large number of sequences that need to be tested to determine which ones can perform a specific function. A strategy is to diversify RNAs known to have the desired property, and assess how many of the resulting sequences conserve this property. Neutral sets of RNA have been explored computationally using secondary structure prediction algorithms[14,15], and experimentally using deep mutational scanning (random mutagenesis combined with a functional assay)[16–24]. The latter studies demonstrated the existence of connected neutral sets in the neighborhood of reference sequences, but remain limited to a few mutations away from it. More recently, machine learning approaches were combined with experimental testing to reach larger diversification, mutagenizing a 20-nucleotide-

[1]Laboratoire de Biophysique et Evolution, UMR CNRS-ESPCI 8231 Chimie Biologie Innovation, ESPCI Paris, Université PSL, Paris, France. [2]Max Planck Institute for Mathematics in the Sciences, Leipzig, Germany. [3]Sorbonne Université, CNRS, Laboratoire de Biologie Computationnelle, Quantitative et Synthétique, UMR 7238, Paris, France. [4]Dipartimento di Fisica, Sapienza Università di Roma, Rome, Italy. [5]Biomolecular Sciences Graduate Programs, Boise State University, Boise, ID, USA. [6]Department of Biological Sciences, Boise State University, Boise, ID, USA. [7]Capital Fund Management, Paris, France. [8]These authors contributed equally: Camille N. Lambert, Vaitea Opuu, Francesco Calvanese. ✉e-mail: philippe.nghe@espci.psl.eu

long region of a ribozyme[25] and reproducing the diversity of naturally occurring ribozymes[26]. However, despite the progress made in the case of protein design[27,28], generative models still perform poorly for larger RNAs that are the size of the autocatalytic ligase and group I introns[29]. Furthermore, predicting catalysis remains difficult in general, and none of these studies have characterized the neutral set for autocatalytic RNA self-reproduction.

Yet, the two autocatalytic RNAs mentioned above have been slightly diversified to create several sequences that collectively reproduce through cross-reactivity networks[30,31]. In these experiments, sequence changes were made primarily to the substrate-binding regions of the ribozyme to change interactions between sequences and cause them to make copies of other sequences more efficiently than their own sequence. However, the majority of the ribozyme sequence outside of the substrate-binding sites remained unchanged in these experiments, and the vast number of sequences capable of these types of reactions remains unknown, limiting our understanding of this type of RNA reproduction. Analysis of evolutionary conservation and structure in naturally occurring self-splicing group I introns (Fig. 1a) suggests a large potential neutral set that has not yet been experimentally evaluated[32]. Here, we devised generative probabilistic models based on statistical learning and structure prediction (Fig. 1b), validated with a high-throughput catalytic assay that serves as a proxy

for autocatalytic reproduction (Fig. 1c, d). This combination of prediction and experimentation enabled the exploration of a very large neutral space of catalytic RNAs derived from Group I Introns, together with estimations of its size. The results demonstrate the potential of this type of generative model and apply it to better understand the vast number of sequences that could contribute to the origin of life.

## Results

### High-throughput assay for self-splicing

To assess the generative power of computational models, we developed a deep sequencing assay, with which we analyzed 24,220 unique RNA sequences (Fig. 1c, d, "High-throughput self-splicing assay" section in Methods). This assay mimics the two steps of the natural self-splicing activity of Group I Introns, from which autocatalytic self-reproduction has been engineered[8,30]. RNA candidates are first synthesized together with a tRNA fragment exon at their 3′ end (yellow in Fig. 1c). The RNA libraries are then incubated together with two substrate RNAs (gray and red in Fig. 1c) at 37 °C. As a proxy for autocatalysis, the RNA is considered active if it transfers its exon to a first substrate, exchanges it for a second substrate, and attaches the extremity of the latter to its own 3′ end (plain red in Fig. 1c). Active RNAs are thus distinguished from inactive ones by deep sequencing, based on the fragment carried at their 3′ end (Fig. 1d). For each variant,

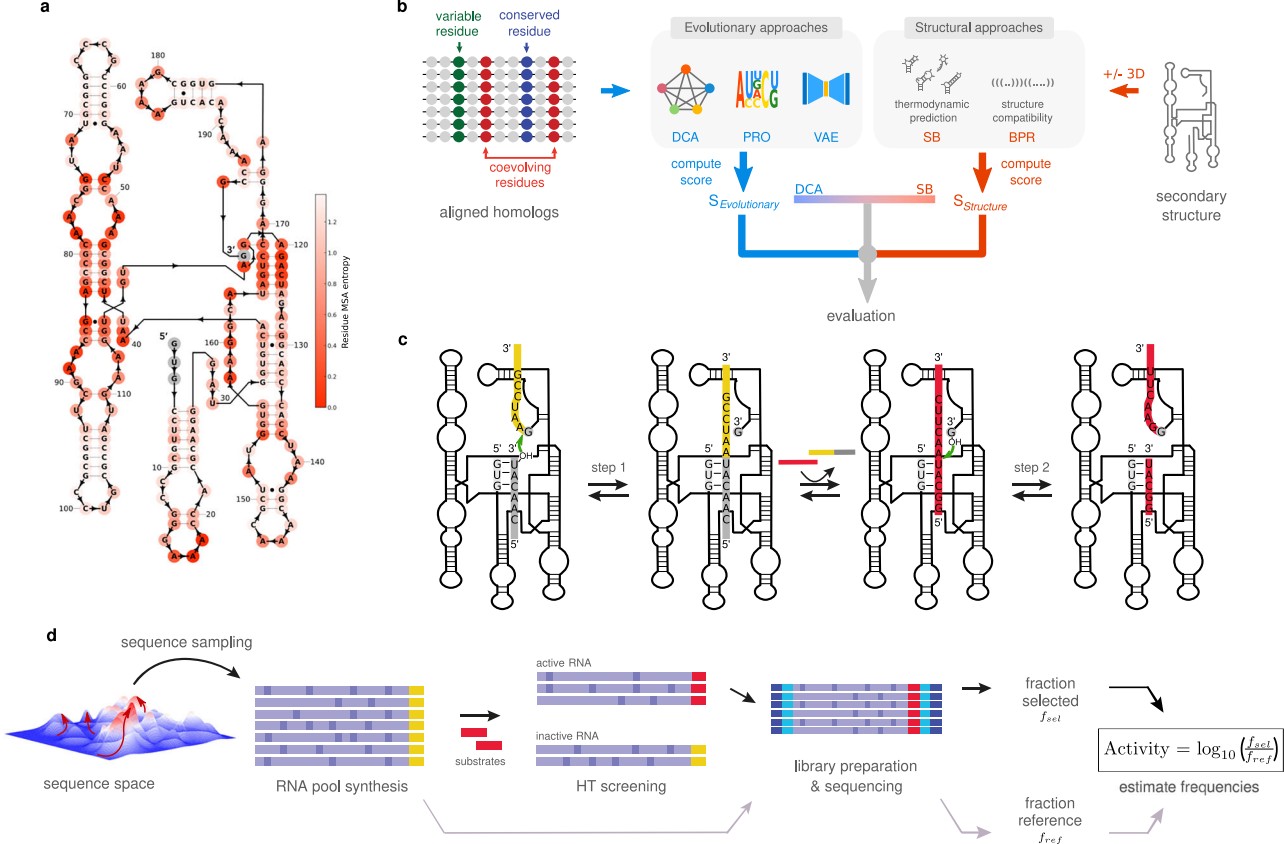

**Fig. 1 | Computational and experimental workflow for the design of artificial group I introns. a** *Azoarcus* ribozyme secondary structure. More conserved positions among homologous sequences of the Multiple Sequence Alignment appear in darker red. **b** RNAs were designed using evolutionary models (DCA, VAE, PRO), structural models (SB, SB-3D, BPR), and a combination thereof (DCA-SB). The input of evolutionary models is a Multiple Sequence Alignment. The input of structural models is ab initio predictions of secondary structure or experimental 3D contacts of Azo. Designs are generated based on probabilistic scores and/or structure scores, depending on the model. **c** Assay mimicking self-splicing: candidate ribozymes (black) are synthesized with an exon sequence (yellow); RNAs are mixed and

incubated with the gray and red substrates; In Step 1, the ribozyme covalently attaches its 3′ yellow to the gray substrate; Then, the product, consisting of the covalently joined gray and yellow sequences, is exchanged for the red substrate; In step 2, the ribozyme ligates the red substrate onto its 3′ extremity. **d** Screening workflow: after computational generation (**b**), the ribozymes are transcribed from a DNA pool, and tested by the screening assay (**c**); after screening, the active ribozymes are amplified with a substrate complementary primer before being sequenced; the frequency of active sequences (carrying S2) post-assay is normalized by the variant frequency pre-assay.

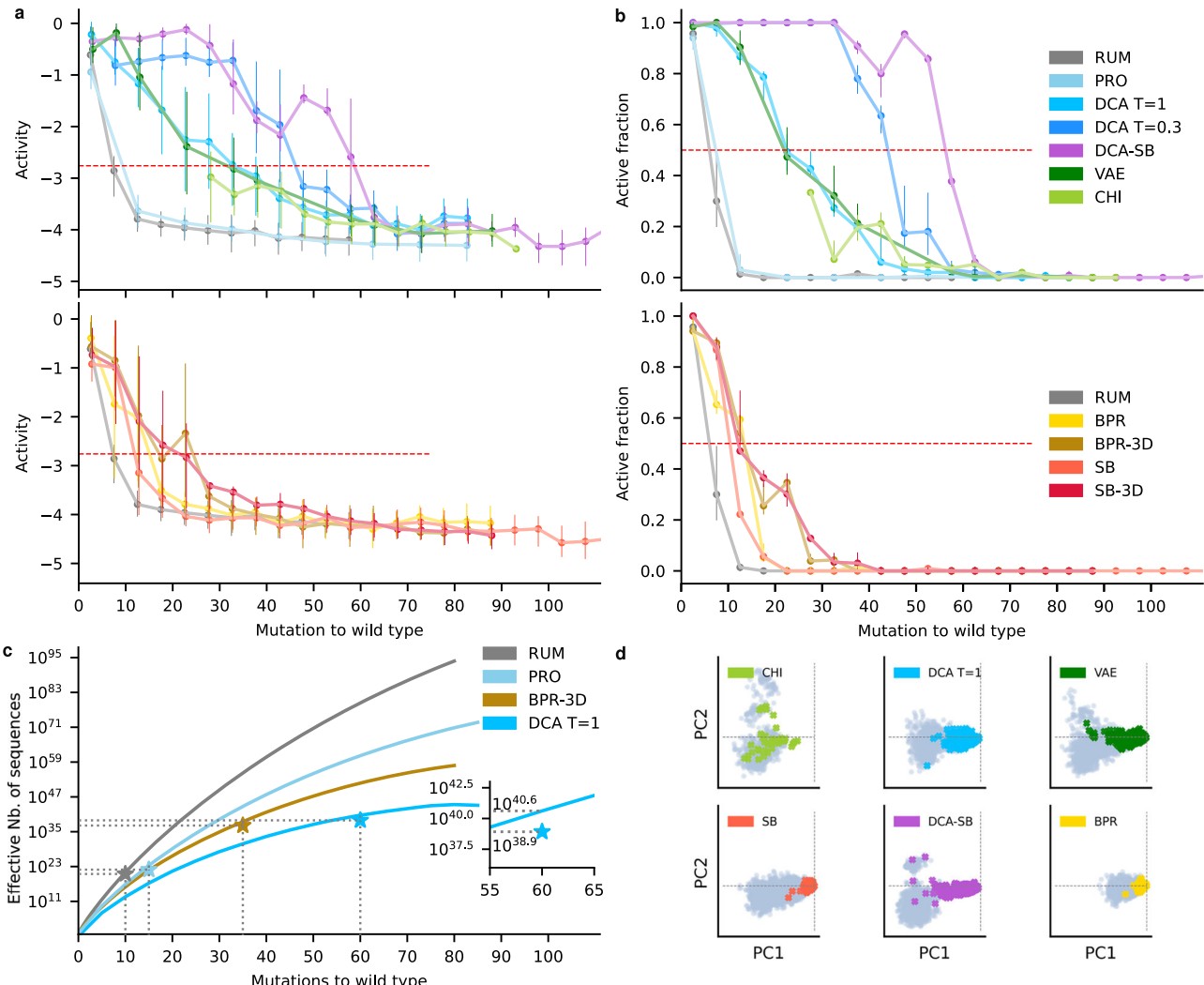

**Fig. 2 | Comparison of the generative power of computational models.**
**a** Experimental activity as a function of the number of mutations relative to Azo, by bins of 5 mutations ($N = 150$ per bin). Top: Statistical learning and hybrid models; Bottom: structure-based models. Per bin, dots are the mean activity, and quartile bars correspond to the first and third quartiles. $N$ per bin per model provided in Source Data file. The red dashed line is the active threshold set at a z-score of 3.09 or equivalently a $p$-value $= 10^{-3}$, which corresponds to an activity of $-2.76$.
**b** Fraction of designs that were catalytically active for each model (Active Fraction) as a function of the number of mutations, by bins of 5 mutations. Dots are the active fraction. Error bars upper (lower) bound is the active fraction including (excluding) activity scores within the 98.5 percentile of the measurement error distribution

around the threshold. $N$ per bin per model is provided in the Source Data file. **c** Effective support size of models as a function of mutational distance, showing how many different sequences the model can generate at any given distance. The star indicates the estimation at $L_{max}$. The star values have been corrected with the experimental active fraction at $L_{max}$, which is of order 1%. Inset: zoom on the experimental correction at $L_{max}$ for DCA $T = 1$. **d** Principal Component Analysis (PCA), with the first two principal components, PC1 and PC2, shown for the generated sequences (gray) overlaid with active sequences (color). All panels are projected on the same axis system, which is the first two principal components of chimeric sequences (analogs of natural Group I introns), with the *Azoarcus* sequence located at the origin. Source data are provided as a Source Data file.

we defined the activity score as the logarithm of the fraction of active sequences after screening, divided by the fraction of sequences before incubation, setting as zero the *Azoarcus* ribozyme reference score (Fig. 2a). The assay was highly reproducible with a Pearson correlation of 0.99 between independent triplicates ($p < 10^{-5}$) (Supplementary Fig. 1). As the reaction consists of attaching fragments to the catalysts themselves via specific binding to substrates, cross-catalysis is expected to be negligible, which is supported by the absence of activity for a large number of mutations (see below). Additionally, we verified *a posteriori* that cross-catalysis negligibly affected the activity score by assaying variants separately by denaturing gel electrophoresis (Supplementary Fig. 2) and sequencing in separate batches (Supplementary Fig. 3), finding that band intensities significantly correlated with sequencing scores (Pearson correlation of 0.61, $p$-value $= 0.008$, $N = 17$). Consistently, mutants that displayed no visible product on the

gel had a sequencing score below the activity threshold (Supplementary Fig. 3). We furthermore tested that incubation in sub-pools yielded the same sequencing scores as within the total pool (Supplementary Fig. 3, "Cross-catalysis tests" section).

We next tested several models by evenly populating bins of 10 mutations up to 150 mutations away from the *Azoarcus* ribozyme (Fig. 2a, 150 sequences or more per model per bin). All models displayed a loss of activity as more mutations were introduced in Azo, with the score reaching a lower plateau corresponding to inactive sequences (Fig. 2a, Supplementary Fig. 4). For random uniform mutagenesis (denoted RUM, see Table 1 for model acronyms), activity was indetectable after 15 mutations, confirming previous studies[21]. The measurement noise distribution was taken as the distribution of scores at more than 100 mutations overall all models (Supplementary Fig. 4), and the threshold for significant activity was set at a z-score of 3.09

**Table 1 | Acronym list of the computational models**

| Acronym | Description |
|---|---|
| RUM | Mutations are introduced by randomly selecting positions and assigning with uniform probability one nucleotide. |
| BPR | Mutations with the Base-Pair Replacement method are introduced following the WT structure: at paired positions, both nucleotides are mutated simultaneously into G–C, G–U, or A–U pairs; otherwise, mutations are random. |
| BPR + 3D | Mutations are introduced as in BPR except positions involved in tertiary interactions (detected by analyzing the tertiary structure), which are kept unchanged. |
| SB | Mutations are introduced with the MCMC algorithm following the Structure-Based (SB) score, which penalizes mispairing under the thermodynamic folding model. |
| SB-3D | Mutations are introduced as in SB except positions involved in tertiary interactions (detected by analyzing the tertiary structure), which are kept unchanged. |
| PRO | Mutations are introduced by randomly selecting positions and assigning nucleotides according to the frequency of the mutation observed in the MSA. |
| DCA $T = 1$ | Mutations are introduced with the MCMC algorithm at $T = 1$ following the DCA model trained on the MSA of homologs, ensuring designs match single and pairwise frequencies observed in the MSA. |
| DCA $T = 0.3$ | Mutations are introduced with the MCMC algorithm at $T = 0.3$ following the DCA model, which concentrates the sampling onto the most likely sequences under the DCA model. |
| DCA-SB | Mutations are introduced with the MCMC algorithm at $T = 0.3$ following the DCA model combined with the SB score, optimizing simultaneously the DCA score and the folding into the WT fold. |
| VAE | Mutations are introduced with the neural network called Variational AutoEncoder (VAE), which is trained on the MSA of homologs. Similarly to DCA, it is trained to maximize the likelihood of the observed MSA. |
| CHI | Chimeric sequences are derived from natural sequences aligned to *Azoarcus* by replacing deletions with *Azoarcus* nucleotides, using only aligned positions to maintain the same length. |

corresponding to an activity score of −2.76 (corresponding to $10^{-2.76} = 0{,}17\%$ of the *Azoarcus* ribozyme activity) and a *p*-value $< 10^{-3}$ for strict positivity ("Activity threshold and active fraction" section). Figure 2b displays the active fraction, defined as the fraction of variants from each model that are above the noise threshold, plotted as a function of the number of mutations introduced relative to the reference *Azoarcus* ribozyme. For each generative model, we report $L_{50}$ as the number of mutations relative to *Azoarcus* that maintains an active fraction of 50% and $L_{max}$ as the number of mutations that maintain 1% (Fig. 2, Supplementary Table 1, 'Definitions of $L_{50}$ and $L_{max}$, significance levels' in Methods). $L_{50}$ can be interpreted as the mutational distance that can be accurately predicted by a given model, beyond which its predictive power rapidly decreases. $L_{max}$ represents the maximum number of mutations that we could reliably assess given our experimental resolution per bin. Note that deeper sequencing of larger RNA pools may allow assessment of functional sequences at frequencies below 1%, which would lead to larger $L_{max}$ values.

**Populating the neutral space with Direct Coupling Analysis**
We first explored the neutral space covered by Group I Introns using Direct Coupling Analysis (DCA), which has been proven to generate in vivo neutral variants of protein enzymes[33]. DCA is a statistical learning approach that accounts for nucleotide conservation and covariation via networks of pairwise couplings[34]. It describes a probability distribution over the space of functional nucleotide sequences. Our model was trained on a Multiple Sequence Alignment of 815 Group I introns sequences that share the same domain composition and secondary structure as the *Azoarcus* ribozyme (Supplementary Fig. 5a). Although sequences of the alignment may differ in length due to deletions or insertions relative to Azo, nucleotide covariations can be learned at positions aligned with the *Azoarcus* ribozyme. Sequences of the same length as the *Azoarcus* ribozyme (197 nucleotides) were sampled by Markov Chain Monte Carlo up to 90 mutations. We used the sequencing-based assay to identify active variants at increasing mutational distance.

We found that the DCA-generated sequences retained activity at higher mutational distances, with $L_{50} = 20$ and $L_{max} = 60$. This was a large improvement as compared to random mutations, for which $L_{50} = 5$ and $L_{max} = 10$ (Fig. 2b, Supplementary Table 1). We also compared the DCA-generated sequences to "chimeric sequences" (CHI), which were generated by adding increasing numbers of mutations to

the *Azoarcus* ribozyme, but only using nucleotide diversity at positions found in other group I introns in our data set, while also removing insertions and deletions to keep a constant length (Supplementary Fig. 5b). These chimeric sequences never reached the 50% success rate per bin (Fig. 2b). Overall, only 6% of them were active, possibly due to the disruption of base-pairs, where this percentage on average corresponds to 7 over the 59 base-pairs found in the *Azoarcus* ribozyme structure. From these comparisons, it is clear that the DCA model can generate active ribozymes at high mutational distances, far beyond what is obvious from random mutagenesis or a naive application of evolutionary conservation.

We next set out to use this DCA model to estimate the size of the neutral set of this type of autocatalytic self-reproducing RNA. It is important to note that the number of potential autocatalytic self-reproducers $\Omega_{DCA}$ predicted from the DCA model cannot be simply calculated by multiplying the total number of possible sequences at mutational distance L by the measured active fraction generated by DCA. Indeed, the DCA model generates predictions focused on a restricted subset of sequences $x$ as compared to random uniform mutagenesis, namely the sequences to which DCA assigns a high probability $P(x)$. Thus, the number of sequences among which the DCA model samples is, in practice, smaller than the total number of possible sequences. On the other hand, to restrict the size of space given the distribution $P$ is not straightforward because any sequence may have a non-zero, however small, probability.

In information theory, the "effective support size" is a term used to define the effective size of a sampling set[35]. The effective support size can be defined for any probability distribution $P$ over all sequences $x$ as $\Omega = \exp(S)$, where $S = -\sum P(x) \log P(x)$ is the Shannon entropy, the sum being taken over all sequences $x$. This definition relies on the following mathematical result: when sampling $N$ sequences $x$ from the probability distribution $P$, the sample probability behaves like $\left(\frac{1}{\Omega}\right)^{N}$ for large enough $N$ (see Supplementary Information for more technical details[35,36]). Intuitively, this means that sampling from the model follows the same statistics as if one were sampling uniformly from a subset of size $\Omega$. In other words, the vast majority of the probability is concentrated in a subset of size $\Omega$, interpreted as the effective number of different sequences that the model can generate.

The full process consists of computing the effective number of sequences $\Omega(L)$ that can be generated by the model (e.g., DCA) at a mutational distance $L$, then multiplying it by the active fraction

experimentally measured at $L$. The maximum of the number at the particular distance L constitutes a lower bound for the whole neutral-set size $\Omega$. In practice, this maximum was determined as the largest $L$ such that we could measure the active fraction to be significantly larger than zero. Although it is not generally known how to compute $\Omega$ for probabilistic or generative models, we devised a semi-analytical method exploiting the structure of DCA (Supplementary Table 2, Supplementary Information "Support size computations")[37], which we applied at each $L$ to obtain $\Omega(L)$. This theoretical number increases sub-exponentially with $L$, reaching $10^{41}$ sequences at 60 mutations (Fig. 2c). The success rate at $L_{max} = 60$ mutations was determined to be larger than 1% with $p < 4.10^{-5}$ confidence. Thus, correcting $\Omega_{DCA}$ with the experimental success rate leads to $\Omega_{DCA} \simeq 10^{39}$.

We next set out to further evaluate the diversity of the sequences generated by DCA. First, we used a Principal Component Analysis (PCA) projection of the DCA-generated sequences and chimeras. We found that the chimeras (representative of natural diversity) clustered near the *Azoarcus* ribozyme in these projections (Fig. 2d), while the DCA-generated sequences bridge the sequence space between the *Azoarcus* ribozyme and natural chimeras, from the perspective of the first two principal components. We note that PC1 strongly correlates with the distance to the *Azoarcus* ribozyme (Pearson $\rho = 0.86$, two-sided $p$-value = $10^{-16}$ corresponding to the numerical precision, under the null hypothesis $\rho = 0$ using the Beta distribution). To further compare the variety of sequences generated by each model, we computed two different mutational distance distributions. First, we computed the distance from each DCA sequence to the closest chimera, including Azo. This can be interpreted as a measurement of how different the samples are from any natural sequences (Fig. 3a, top). Second, we computed the distances between all pairs of sequences within each model (Fig. 3a, bottom). Experimentally active DCA sequences were found to be on average 25 mutations away from each other (Fig. 3a, bottom), on average 18 mutations away from any chimera, and up to 55 mutations away from any chimera (Fig. 3a, top, Supplementary Table 3). These distances were comparable to those found between all chimeras (active and non-active), which had 32 mutations between them on average and up to 59 mutations away from each other (Fig. 3a, Supplementary Table 3). It thus appears that DCA achieves a form of interpolation, by generating a diversity of experimentally active sequences similar to the diversity of natural Group I introns (Fig. 3b, c), as well as a form of extrapolation, by generating active sequences as far away from any chimeras as the latter are from each other. Extrapolation is further confirmed by the largest distance between active DCA designs and chimeras being 99 mutations (Supplementary Table 3), well beyond the largest distance between any two chimeras.

### Generative power of statistical and structural methods

These initial results suggest a trade-off between extensively exploring all regions of genotype space and increasing the probability of successfully finding active sequences. For instance, random mutagenesis considers all possible mutations and would theoretically yield the highest possible $\Omega$, but its success rate decreases very quickly below our ability to experimentally detect active fractions. In general, a less restricted model should eventually discover a larger total number of active sequences, but with a much lower success rate, leading to inefficient experiments. In contrast, more constrained models like DCA generate functional sequences with a higher number of mutations, but at the cost of exploring more limited regions of the sequence space. This further suggests that for a given model, different choices could be made to fine tune the model to achieve a desired outcome, such as identifying active sequences with a larger number of mutations (as quantified by $L_{50}$ and $L_{max}$), or to yield a larger effective number of generated sequences at a given distance (as quantified by $\Omega(L)$). Overall, the estimation of the number of active sequences by calculating effective support size is not obvious because, for any given

model, it depends on how $\Omega$ increases with the number of mutations as well as on the $L_{max}$ that can be validated experimentally.

To explore this trade-off, we tested more relaxed models, which we expected to access a larger diversity but at a lower mutational distance. The profile model (PRO) was a randomization approach where each nucleotide was replaced according to its position-specific frequency in the Multiple Sequence Alignment. It is less constrained compared to DCA, but still more constrained than random (RUM) because it utilizes some evolutionary information from nucleotide conservation. This PRO model was only marginally more accurate than RUM, with $L_{50} = 5$ (Fig. 2b, Supplementary Table 1), $L_{max} = 15$, and $\Omega_{PRO} = 10^{22}$. Recall that the success rate of random mutagenesis (RUM) was nearly identical to this with $L_{50} = 5$, $L_{max} = 10$, and $\Omega_{RUM} = 10^{20}$, and these metrics were much lower for both models than for DCA (Fig. 2b, Supplementary Table 1). Limitations of the profile model likely result from a high probability of disrupting the secondary structure when selecting nucleotides independently, as 60% of *Azoarcus* ribozyme positions are base-paired (Fig. 1a). To assess the impact of base pairing, we tested the random replacement of base pairs of *Azoarcus* ribozyme with other canonical pairs, combined with randomized nucleotides at unpaired positions, which we termed the Base-Pair Replacement model (BPR). This model performed significantly better than the profile model, with $L_{50} = 15$, $L_{max} = 20$, and $\Omega_{BPR} = 10^{29}$. Further constraining the model by forbidding variations at tertiary contacts (30 positions in Azo, Supplementary Fig. 6a, BPR-3D model) better populated the tail of the distribution of active RNAs, with $L_{50} = 15$ and $L_{max} = 35$. Ultimately, $\Omega_{BPR-3D} = 10^{37}$, which was still smaller but nearing $\Omega_{DCA}$.

Because the base-pair replacement model (BPR) improved prediction success at higher mutational distances, we next explored the impact of using secondary structure prediction. We generated sequences that were predicted to form the correct secondary structure from a thermodynamic model (SB model for 'structure-based')[38–40], using a heuristic that retained the pseudo-knot as a probabilistic signature in the contact map (Supplementary Fig. 6e)[41–43]. This model performed only slightly better than RUM, with $L_{50} = 10$ and $L_{max} = 20$. This limited performance could be due to discrepancies between the predicted and the actual structure (Supplementary Fig. 6c) or to structure energy minimization being non-optimal for catalysis (Supplementary Fig. 7). Fixing positions involved in 3D contacts[44] (SB-3D model) again improved the performance at higher distances, with $L_{50} = 10$ and $L_{max} = 40$. The higher $L_{max}$ was beyond that of the BPR-3D model, suggesting that cooperative effects that determine structure stability in the SB model may make a cumulative difference, but only when critical tertiary interactions are unperturbed.

Despite the success of the DCA model, we considered that pairwise statistical covariation alone might miss the importance of higher-order interactions of three or more nucleotides. Such higher-order interactions are known to be important for group I intron ribozyme catalysis. For example, there are several base-triples in the "G-binding pocket" of the active site[45]. To further explore more constrained models, we used Variational AutoEncoders[46] (VAE) to encode higher-order interactions in a neural network and generated sequences from this model. Overall, VAE covered a similar diversity to DCA as shown by PCA projections (Fig. 2d) and only a slightly lower capacity for mutagenesis, with $L_{50} = 15$ and $L_{max} = 60$. The absence of improvement compared to DCA may be explained by the modest number of training sequences ($N = 816$), which might be insufficient to resolve higher-order couplings[47].

We next set out to further improve the DCA-based models by introducing some constraints. We sampled sequences with higher probability according to DCA by lowering a parameter termed "sampling temperature", which controls the degree of random diversification during Monte Carlo sampling, from $T = 1$ to $T = 0.3$. This alone showed a dramatic improvement in model success at higher

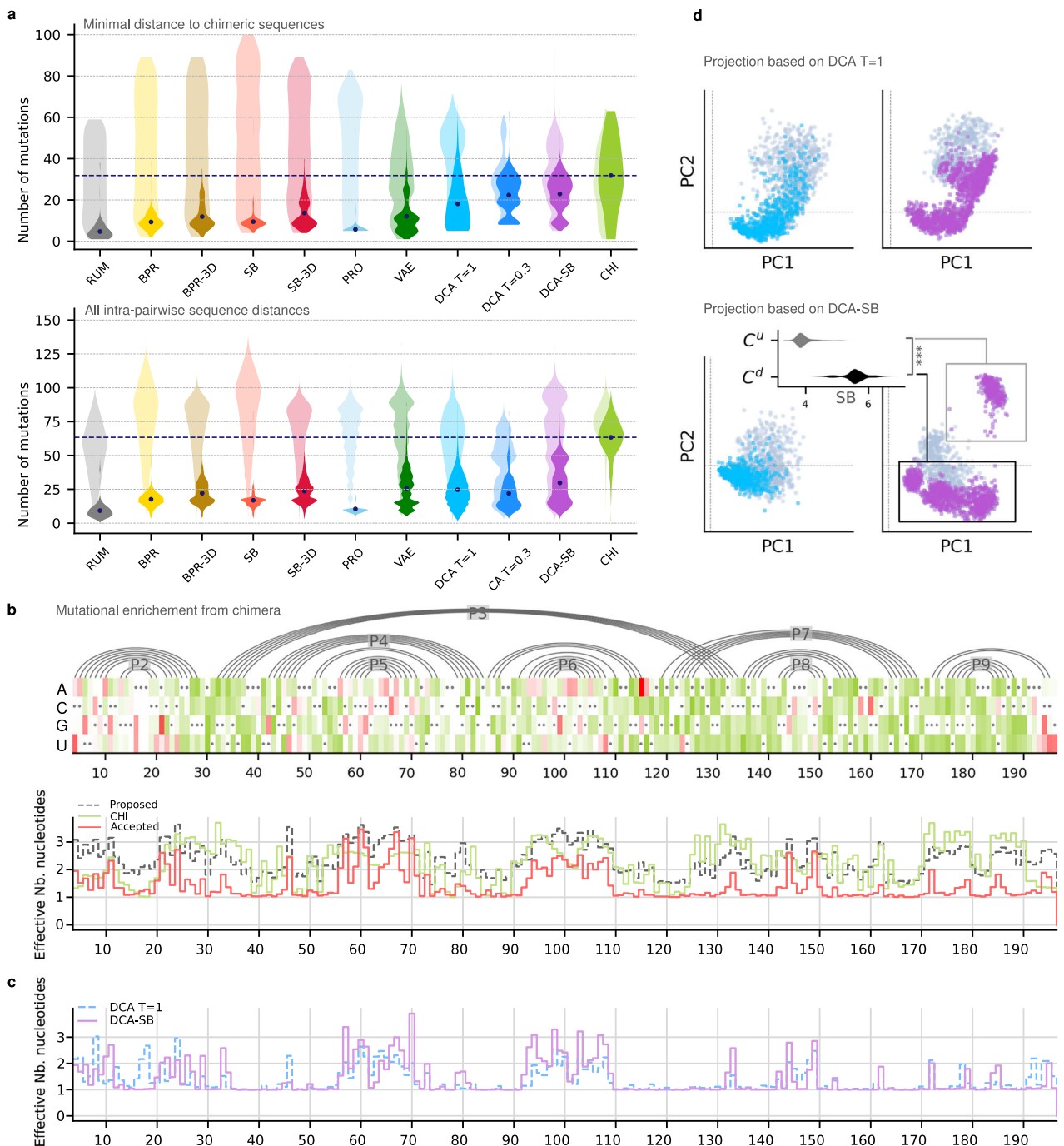

**Fig. 3 | Diversity explored by the generated sequences. a** Upper panel: Violin plots of minimal distance to the set of chimeric sequences for generated sequences (light shades) and active sequences (solid shades), per model, with identical colors as in Fig. 2b; The average of distances for chimeric sequences is represented by a dashed line. Lower panel: Violin plot of distances between all pairs for generated sequences (light shades) and active sequences (solid shades), per model. **b** Mutational enrichment from chimeric sequences. Top: At each position (x-axis), *Azoarcus* ribozyme nucleotides are marked by a black dot, and shades are the log ratio of mutation frequencies relative to *Azoarcus* ribozyme in chimeras (green) and in active variants over all models (red). Bottom: effective number of nucleotides in

chimera (green), candidate designs (black), and active designs (red). **c** Mutational enrichment due to the structure: Effective number of nucleotides per position in DCA $T = 1$ (blue) and DCA-SB (purple). **d** Top: PCA projection of DCA (left) and DCA-SB sequences (right), on the 2 principal components of the DCA $T = 1$ set. Active sequences are colored and inactive ones are gray. Bottom: same but in the principal components of the DCA-SB set, revealing an upper cluster ($C^u$) populated by DCA-SB only, distinct from the cluster below ($C^d$) that comprises all DCA sequences. Inset: Distribution of structure scores in the two clusters, showing that $C^u$ has a significantly better structure score (***, one-sided *t*-test, *p*-value = $10^{-16}$, numerical precision). Source data are provided as a Source Data file.

mutational distances with $L_{50} = 45$ and $L_{max} = 60$. In a separate approach, we combined the simple secondary structure maintaining approach (SB) with DCA to generate the DCA-SB model, where DCA is sampled at $T = 0.3$. This model also showed a very high success rate at higher mutational distances, with $L_{50} = 55$ and $L_{max} = 65$, with a

significantly higher success rate in the range 60–70 compared to DCA alone (two-sample *t*-test, $p < 10^{-5}$). Interestingly, the DCA-SB sequences introduced diversity in the loops P5 and P6 (Figs. 1a, 3c), a diversity that was not found in DCA alone. The better exploration by DCA-SB was confirmed by PCA projections, where the DCA-SB sequences covered

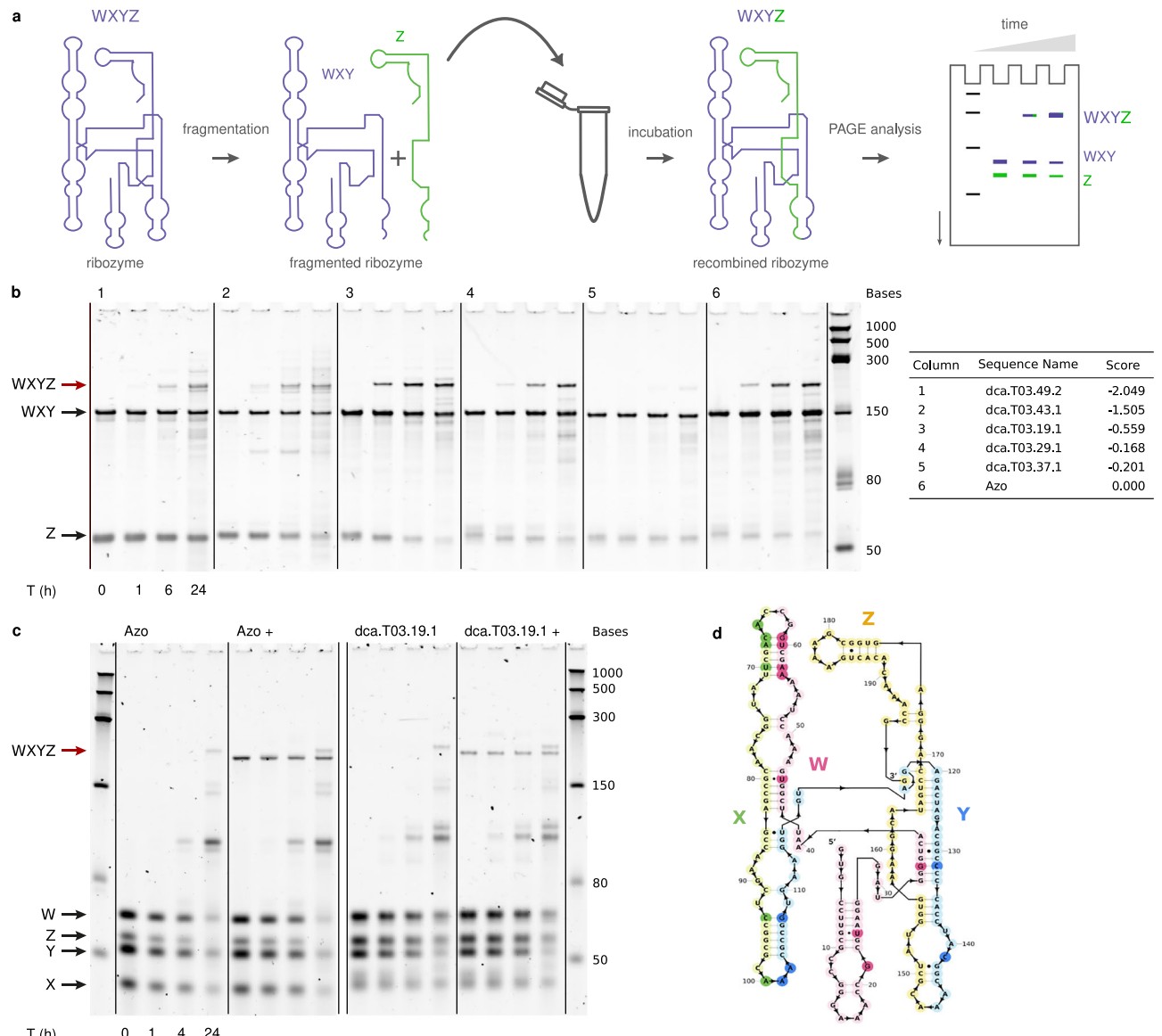

| Column | Sequence Name | Score |
|---|---|---|
| 1 | dca.T03.49.2 | -2.049 |
| 2 | dca.T03.43.1 | -1.505 |
| 3 | dca.T03.19.1 | -0.559 |
| 4 | dca.T03.29.1 | -0.168 |
| 5 | dca.T03.37.1 | -0.201 |
| 6 | Azo | 0.000 |

**Fig. 4 | Evidence for self-reproduction. a** Two-fragment self-reproduction experimental assay. The ribozymes are first fragmented into two pieces (WXY and Z) at position 145, then incubated at defined time points. The formation of longer covalent copies is monitored by gel electrophoresis. **b** Two-fragment self-reproduction assay for 5 designs and Azo, assayed for 0, 1, 6 and 24 h. The assayed ribozymes are documented in the table on the right of the gel: the number of mutations from the *Azoarcus* ribozyme is indicated in their name, and their activity score for the self-splicing assay in the last column. The molecular scale in nucleotides. The red arrows are pointing at the recombined covalent ribozymes. For other designed ribozymes with 15–60 mutations, see Supplementary Fig. 8. **c** Four-fragment self-reproduction assay for one design and Azo, assayed for 1, 2, 4 and 24 h. The dca.T03.19.1 design and *Azoarcus* ribozyme are first fragmented into four fragments (W, X, Y and Z) at the positions 62, 99 and 145. The molecular scale in nucleotides. Two conditions were assayed: the four fragments in the absence of the covalent ribozyme; and the four fragments in the presence of the covalent ribozyme (condition +). The red arrow shows the presence of the fully recombined covalent ribozymes. **d** The dca.T03.19.1 ribozyme. The colors represent the 4 different fragments, and the nucleotides colored with a darker shade highlight the position of mutations compared to the *Azoarcus* ribozyme.

the DCA cluster plus an additional cluster that was characterized by a better structure score (Fig. 3d).

Focusing on the models that identified activity at higher mutational distances than the standard DCA, namely DCA $T = 0.3$ and DCA-SB, we computed the support size for these models over a limited range of mutational distances (Supplementary Table 2). Over the explored range of mutations, $\Omega_{DCA\ T=0.3}$ and $\Omega_{DCA-SB}$ were found to be many orders of magnitude below $\Omega_{DCA}$, consistent with stronger design constraints (Supplementary Table 2). This result illustrates how models better for introducing many mutations at once are also more constrained, resulting in a reduction in the total effective number of sequences that can be discovered.

## Robustness and generality of the approach

We have used a two-step splicing assay as a proxy for the self-reproduction from fragments because it uses the same catalytic mechanism. To confirm that the DCA-generated sequences can, in fact, self-reproduce from fragments, we chose individual sequences at varying distances from the *Azoarcus* ribozyme up to 60 mutations (Supplementary Material 'Self-reproduction assay'). We split sequences into either two or four RNA fragments, allowed them to react, and monitored the appearance of the full-length ribozyme (Fig. 4, Supplementary Fig. 8)[8]. More than 60% of the sequences ($N = 17$) that had a score larger than the activity threshold were found to covalently self-reproduce from fragments. Importantly, we observed self-

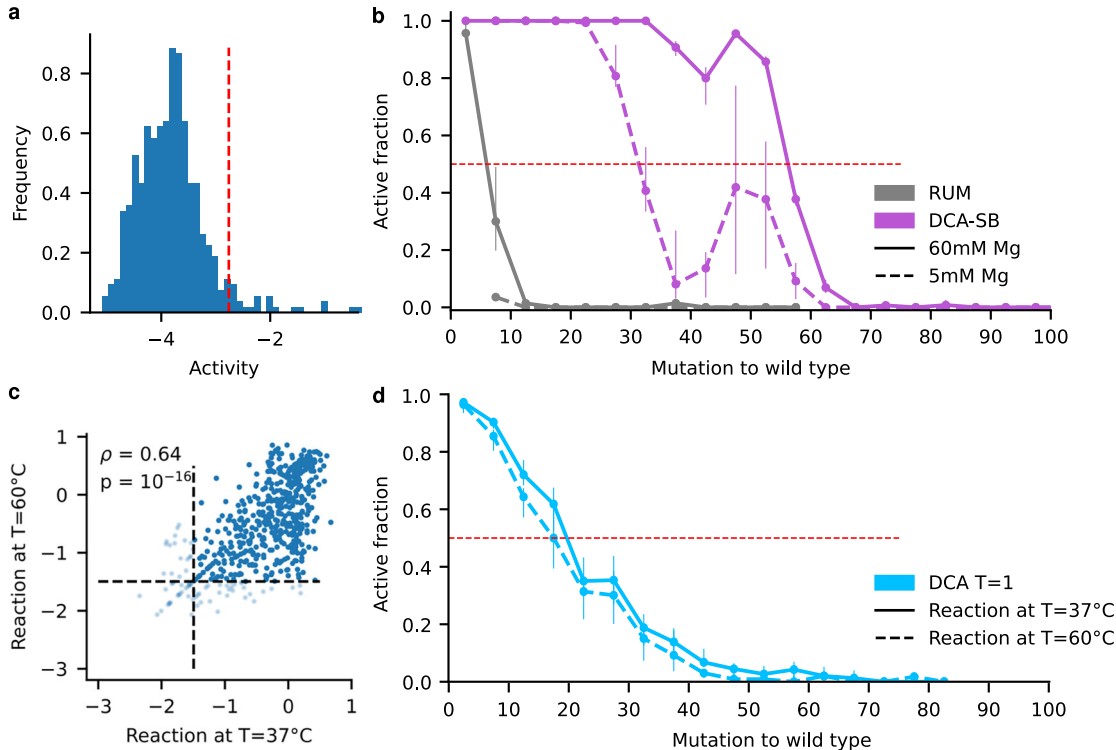

**Fig. 5 | Robustness and generality. a** Designed DCA sequences at 60 mutations from the WT. The histogram shows the activity scores of 991 detected designs (out of 1000 designs) with 22 active sequences beyond the activity threshold (equal to −2.76, red dotted line), corresponding to 2% of the pool. **b** Comparison of active fraction as a function of the number of mutations for the RUM and DCA-SB models, for different $MgCl_2$ concentrations: 60 mM as our standard condition (solid lines) and a lower concentration of 5 mM (dashed lines). $N$ per bin per model provided in the Source Data file. Dots are the active fraction. Error bars upper (lower) bound is the active fraction including (excluding) activity scores within the 98.5 percentile of the measurement error distribution around the threshold. **c** Sequencing score at 37 °C versus 60 °C for the same DCA pool. Due to lower sequencing depth as compared to assays of Fig. 2, a more stringent activity threshold of −1.5 was used (see "Methods" section). The Pearson correlation is 0.64, two-sided $p$-value = $10^{-16}$

(numerical precision). $N$ = 558, comprising 484 active for both conditions (upper-right quadrant), 25 inactive for both (lower-left), 19 active at 60 °C but not 37 °C, 30 active at 37 °C but not 60 °C. **d** Active fraction as a function of the number of mutations for the DCA model at 37 °C (plain line) and 60 °C (dashed line). Sequences with a score below the activity threshold in at least one condition are in light color. Note that the fraction is computed over all designed sequences, including those that were not active enough to be detected by sequencing (thus counted as inactive). $N$ per bin per model provided in the Source Data file. Dots are the active fraction. Error bars upper (lower) bound of vertical bars is the active fraction including (excluding) activity scores within the 98.5 percentile of the measurement error distribution around the threshold. Source data are provided as a Source Data file.

reproduction for DCA sequences with up to 60 mutational differences from the *Azoarcus* ribozyme. We note that the self-reproduction reaction can be sensitive to the structural context of the CAU tag sequences, and many more sequences might be able to self-reproduce with minor changes to the precise tag location. These results confirm that a majority of catalytically active group I intron ribozymes artificially generated by DCA are capable of covalent self-reproduction from fragments.

We next thought to strengthen our estimation of the size of the neutral set obtained with the DCA model. As mentioned, the estimation is based on the support size curve combined with the number of active sequences found at different mutational distances (Fig. 2), and is sensitive to deviations at the largest mutational distances. To better establish our estimation of $\Omega_{DCA}$, we generated a DCA pool of 1000 sequences at 60 mutations and experimentally determined the active fraction. The results showed that 2.2% of the sequences were functional, consistent with our initial estimate. The larger sample size enhances the level of significance, leading to a $p$-value < $10^{-21}$ for having active sequences in the 60-mutation bin (Fig. 5a).

To relate our findings to possible prebiotic scenarios, we assessed the impact of environmental conditions, magnesium concentration, and temperature, which are both known to affect structural stability and catalysis in RNA. While the experiments reported so far were done at 60 mM magnesium concentration, we used a 5 mM magnesium

concentration to run the assay for the pools of the RUM and DCA-SB, the models that introduced the least and the most mutations, respectively. As expected, the active fraction decreased in both cases, consistently in each mutation bin (Fig. 5b). The impact on the RUM was dramatic, with $L_{50} = 0$ and $L_{max} = 5$, due to initially low metrics. However, the maximum diversification was found to be robust for the DCA-SB model, with $L_{50} = 30$ and $L_{max} = 60$.

Regarding the effect of temperature, the *Azoarcus* ribozyme strongly degrades above 70 °C (half-life shorter than 30 min). We thus tested the standard DCA model pool at 60 °C, below this upper limit, but well above the 37 °C of the initial test. Sequencing scores correlated well between the 2 conditions ($r = 0.64$, $p$-value < $10^{-5}$, Fig. 5c). The region below the x = y diagonal appears more populated than above it (Fig. 5c), confirming the expectation that a given sequence should have a better activity at 37 °C than at 60 °C. Qualitatively, a large majority of the mutants detected by sequencing were found to be above the activity threshold for both temperatures (484 over 558 detected sequences, those below the threshold or non-detected being considered inactive). Consistently, the active fraction as a function of the number of mutations was slightly lower at 60 °C than 37 °C, yet both temperatures followed the same trend (Fig. 5d), leading to $L_{50} = 15$ and $L_{max} = 45$ at 60 °C (compared to $L_{50} = 20$ and $L_{max} = 60$ at 37 °C).

Finally, we wondered if the results were specific to the *Azoarcus* ribozyme model system, or if it could be extrapolated to other

ribozymes. First, we applied the DCA model using a group I intron from *Phormidium* sp. as a starting point for mutagenesis (Supplementary Fig. 9, Supplementary Information 'Phormidium'). We measured $L_{max} = 25$ for the DCA model and $L_{max} = 35$ for the DCA-SB model. Although the degree of diversification was lower than for *Azoarcus*, this indicates that the diversification approach can be generalized to extend the neutral set from other starting points. The next question is whether these findings might be applied to ribozymes that differ by their activity, structure, or length. A recent analysis shows that the effective number of RNAs with a given activity within a family of structural analogs can be estimated, according to DCA, using a power law $d^L$, where $L$ is the length of the RNA and d is an effective nucleotide diversity per position found to be 1.74[37]. Following this analysis, we computationally examined three other ribozyme families with diverse lengths: Hammerhead (66 nucleotides), glmS (167 nucleotides), and Bacterial RNase P class A (367 nucleotides). We observed comparable trend of the support size versus number of mutations, which furthermore matched the scaling law of support size versus length formerly found on a large number of naturally occurring RNAs[37] (Supplementary Fig. 10). These results confirm the generality of generative approaches for functional RNAs, also found by a deep learning method[26], and indicates a rule of thumb to extrapolate our DCA-based estimation of diversity to other ribozymes.

## Discussion

The mapping of molecular function to sequence space is important for understanding many biological questions, ranging from biotechnology applications to the origin of life. Sequence space increases exponentially with the length of a genetic sequence (DNA, RNA, or protein), and this presents several challenges for studying larger RNA molecules that are important models for origins of life research. For the 197-nucleotide-long *Azoarcus* group I intron RNA studied here, there exist $10^{118}$ possible sequences that make up the sequence space. It is not possible to synthesize all of these RNA sequences, and we must therefore develop methods to maximize information gained from experiments. Here, we demonstrated an approach that allows an efficient exploration of this sequence space to map genotypes capable of an autocatalytic self-reproduction reaction. By comparing several statistical models and reporting the consistency of their predictive power over thousands of measurements each, we provided a method for future applications of this approach to other systems.

Based on the best-performing DCA model and the measured activity of randomly sampled sequences, we suggest that there exist over $10^{39}$ autocatalytic self-reproducers that make up the neutral set for this function. This estimation is based on our experimental detection limit and therefore represents a lower bound. While this is a lower bound of potentially autocatalytic sequences, it provides a quantitative evaluation of the sequence space to guide future questions and research. At the moment, the frequency of other ribozyme activities may be estimated from randomization techniques. Ligase ribozymes, from which replicases were derived, were selected from random pools of size $10^{15}$ sequences[48]. The diversification potential of aminoacylation ribozymes, relevant to the origin of translation, was explored by mutagenesis in a subregion of length 21 nucleotides, showing that $10^5$ sequences were active among the $4.10^{12}$ possible ones[22]. In general, these studies and the current work suggest that ribozymes are abundant in sequence space, but suggest that different structures and functions likely have different frequencies.

The Direct Coupling Analysis approach can be applied to additional RNA systems to further advance origins of life research or other RNA research. We found that diversification was equally efficient when starting from different group I intron ribozymes (*Azoarcus* and Phormidium ribozymes). Smaller group I intron analogs are also known to exist, such as a 140-nucleotide-long ribozyme derived from a shortened SunY ribozyme. In the long term, generating RNAs with the same activity but much less sequence identity or without length constraints should be possible, and has been demonstrated in proteins[49]. Beyond group I intron analogs, relaxing the structural constraints would further expand the range of RNAs capable of self-reproducing. For instance, the family of group II intron ribozymes is structurally distinct but also catalyzes splicing reactions between RNAs. Further, smaller and simpler instances of autocatalytic RNAs exist, but have not been extensively characterized experimentally. For instance, engineered autocatalytic ligase ribozymes are 67 nucleotides long, overall relying on an enzyme-substrate complex of 134 nucleotides[50]. Even smaller ribozymes were recently found to emerge from random mixtures of activated oligomers, with lengths as short as 20 nucleotides[51], relying on 40-nucleotide complexes. By studying reproducing ribozymes of different sizes and structures, it should be possible to determine scaling laws relating ribozyme length and diversity, and allow estimates of the size and connectedness of the neutral sets[37]. We started with only 815 unique sequences to train our models, suggesting that any RNA structure could be similarly studied with only minimal known natural examples, or by generating a good initial data set experimentally. We note that the choice of training data is important, but the optimal criteria might change for different sequences and structures. The methods for selecting training data remain an active area of research in itself[52].

Autocatalytic RNAs should also be considered in the larger picture of prebiotic environments, which would likely alter the activity of certain sequences and impact the mapping of genotype to function[53]. Certain prebiotic processes could have actually facilitated the production or preservation of autocatalytic RNA. For instance, interactions with the mineral surfaces can favor structured RNAs that are prone to catalysis[54]. Peptide-RNA interactions have been shown to form coacervate structures that can improve RNA functions[55,56]. On the other hand, the prebiotic milieu could be constraining. In particular, the concentrations of different metal ions can impact ribozyme activity. Lower magnesium concentration tends to reduce the fraction of active molecules, although we found that magnesium concentration in the millimolar range had a limited impact on our identification of active sequences.

Generative approaches, as developed in this work, could uncover the full extent of RNA sequences capable of achieving functions relevant to the RNA world. The highly active developments in artificial intelligence, including large language models, for RNA structure[29,57,58] and function[59,60] hold the promise to systematically design RNAs and ultimately reveal the variety of accessible pathways for prebiotic evolution.

## Methods

### High-throughput self-splicing assay

The assay aims at discriminating against an activity similar to self-splicing, the reaction catalyzed by the wild-type *Azoarcus* GII. For the natural self-splicing reaction, the GII binds a free GTP, allowing the attack of A. Then, the 3′ of A attacks the 3′ of the GII to allow the formation of a covalent bond between the two exons, while releasing the GII.

To mimic this self-splicing mechanism, we devised an experimental assay comprising two steps, as shown in Fig. 1c. All ribozymes were produced with a 15-nt-long sequence at their 3′ extremity, called sequence A (yellow region in Fig. 1c), which corresponds to the wild-type exon of *Azoarcus* GII. First, the sequence A is transferred at the end of the substrate called S1 (grey region in Fig. 1c). Then, a second substrate called B-S2 (red substrate in Fig. 1c) binds into the recognition site of the RNA, leading to the transfer of a cleaved section S2 of B-S2 at the 3′ end of the ribozyme and the release of a smaller B fragment (fraction of the red substrate that remains attached to the 5′ GUG extremity of the ribozyme, right-most panel of Fig. 1c). The ribozymes with the sequence S2 (fraction of the red substrate attached to the 3′

extremity of the ribozyme, right-most panel of Fig. 1c), thus considered active, were selected during the later steps.

We performed three large pooled experiments, wherein we assayed thousands of sequences simultaneously. For each computationally designed pool of RNA sequences, we ordered the corresponding DNA templates with the exon at the 3′ end and the T7 promoter in the 5′ end. The single-strand DNA oligo pools containing 12,000 or 18,000 sequences were purchased from Twist Bioscience. We amplified the DNA pools by PCR (-15 cycles) with the KAPA HiFi HotStart ReadyMix (Roche), and purified the samples with the NucleoSpin Gel and PCR cleanup (Macherey Nagel). The DNA templates were then transcribed using the HiScribe T7 High Yield RNA Synthesis Kit (New England Biolabs) for 4 h at 37 °C to produce RNA molecules. The samples were then subjected to phenol-chloroform extraction and ethanol precipitation using 0.1 volume of 3 M Sodium Acetate (Sigma) and 2.5 volumes of cold 100% ethanol. After extraction, the samples were treated with the DNAse I (New England Biolabs) and PAGE purified on an 8% urea PAGE.

For each designed RNA pool, we performed two sub-experiments. The first sub-experiment is the self-splicing assay. The second one is a control experiment to correct for the biases of the relative quantity of each synthesized ribozyme within the corresponding pool (before reaction), used to compute the activity scores. For the self-splicing assay, 2 $\mu$M of ribozymes were incubated with 25 $\mu$M of substrates S1 and B-S2 in a buffer (30 mM EPPS, pH 7.5, 60 mM MgCl2) at 37 °C in a final volume of 20 $\mu$L. Two samples were taken during the incubation, at 0 and 60 min, mixed with loading solution (70% formamide, 130 mM EDTA, 0.1% xylene cyanol, 0.1% bromophenol blue), and loaded on an 8% Urea PAGE. The reaction was quenched by adding 60 mM of EDTA, and the ribozymes were cleaned using the Monarch RNA cleanup kit (New England Biolabs) with an adjusted volume of ethanol and binding buffer. For the control experiment, no substrate was used, and the ribozyme pool was directly cleaned with the Monarch RNA cleanup kit.

The RNA samples were then prepared for sequencing, using the NEBNext Ultra II Directional RNA Library Prep Kit for Illumina (New England Biolabs). For the self-splicing experiment, the primer used during the reverse transcription corresponds to the sequence complementary to the S2 substrate, whereas the sequence complementary to sequence A was used for the control experiment. The samples were sequenced on a NovaSeq SP flow cell (2*250 nt, 2*800 M reads) in paired ends and with 25% of PhiX by the NGS platform at Institut du Cerveau et de la Moelle épinière (ICM, Paris).

To test the robustness of our experimental protocol across the 3 pools, we introduced a common set of 355 sequences, which showed a very consistent activity computation across all pools ($\rho$ = 0.99 correlation between each pair of pools, Supplementary Fig. 1).

## Cross-catalysis tests

Cross-catalysis within a pool would result in false positives (apparently active ribozymes that are in reality inactive), which in turn could lead to overestimation of the lower bound on the number of ribozymes. To test for this, we performed a series of experiments ensuring that cross-catalysis was negligible.

First, we assayed 24 sequences by denaturing gel electrophoresis, individually (Supplementary Fig. 2). We used identical reaction conditions as the pooled high-throughput assay: 2 $\mu$M of ribozymes incubated with 25 $\mu$M substrates at 37 °C for 1 h. These sequences were selected among the pool designed by DCA to cover the range of mutations 15–60 from Azo, with various scores of the assay as measured by sequencing. Overall, all the sequences that had a sequencing score above the threshold also displayed a form of catalytic activity visible on the gel. 1 in the 24 sequences was a false negative according to sequencing: it displayed products on the gel but a sequencing score below the threshold. However, neither false negatives impact our lower bound estimation of the number of ribozymes, nor do they indicate the presence of cross-catalysis as the latter results in false positives.

Second, we measured by the sequencing assay the activity score of a set of sequences spanning a range of activity scores, comparing the score obtained in a pool and as separate individual measurements. The sequences were produced, purified, and tested separately. The scores reproduced well the relative experimental activity measured in the pool, with $\rho$ = 0.95 with $p$-value < $10^{-5}$ (Supplementary Fig. 3b).

We further tested that active ribozymes did not affect the measured activity of each other within a pool as a function of their relative concentration, by taking a subset of 12 only active sequences, tested as a subpool. The activity score of this subset incubated as a separate, smaller pool correlated strongly with the scores within the total pool, with $\rho$ = 0.98 with $p$-value < $10^{-5}$ (Supplementary Fig. 3c), showing that the overall concentration of active variants did not affect the activity score.

## Self-reproduction assay

The fragmentation strategy previously reported for fragmenting the *Azoarcus* ribozyme[8] was generalized to the artificial ribozymes. We fragmented the ribozymes at the nucleotide positions 145–147, which we referred to as the Y/Z junction. This fragmentation site is located in the loop of the P8 paired region (Fig. 4a). For each ribozyme, two fragments were produced with the addition of short sequence fragments necessary for the recombination reaction to occur. For the first fragment (WXY fragment), a 'CAU' tag was inserted at the 3′-end, and for the second fragment (Z fragment), a 'GGCAU' tag was inserted at the 5′-end by PCR.

The self-reproduction assays were carried out as follows. 1 $\mu$M of each ribozyme fragment was incubated in a buffer (30 mM EPPS, pH 7.5, 60 mM MgCl2) at 37 °C. Samples were taken at different time points during the incubation, mixed with loading solution (70% formamide, 130 mM EDTA, 0.1% xylene cyanol, 0.1% bromophenol blue), and loaded on an 8% Urea PAGE. The gels were stained with SyBr Gold (ThermoFisher) diluted in TBE buffer.

To fragment a ribozyme into 4 fragments[8] (W, X, Y Z fragments, Fig. 4d), the 'CAU' tag was inserted at the 3′-end of the fragments W, X and Y and the 'GGCAU' tag was inserted at the 5′-end of the fragments X, Y and Z. Similarly, the self-reproduction reaction was performed by incubating 1 $\mu$M of each 4 fragments in the incubation buffer (30 mM EPPS, pH 7.5, 60 mM MgCl2) at 37 °C. The course of the reaction was followed by analyzing samples on an 8% Urea PAGE.

## Construction of the MSA

To build the MSA, we used the wild-type sequence (197 nucleotides long) of *Azoarcus* GI Intron and its known secondary structure derived from the PDB X-ray structure 1U6B[61] in order to detect homologous sequences from the RFAM database[62]. To account for similarity but also secondary structure compatibility, we built a seed alignment containing only the *Azoarcus* sequence and its secondary structure, which we fed to the alignment algorithm implemented in the package Infernal (version 1.1.4)[63] to search for homologs in the 2611 sequences of the RF00028 RFAM family. We filtered out all the sequences for which the computed e-value is >$10^{-3}$, which is the probability of finding the obtained alignment score randomly. 816 sequences were found below the homology threshold of $10^{-3}$ and aligned using Infernal. Supplementary Fig. 5a shows the diversity per position in this alignment, measured as the exponential of Shannon entropy, and the frequency of gaps per nucleotide position.

## Calculation of activities from sequencing results

To estimate the experimental activities, we computed the frequencies of designed sequences in: (i) the reference condition before the catalytic reaction, and (ii) the reacted condition where the substrate has been mixed with the designs and then incubated. For both conditions,

we mapped each paired-end read to the closest designed sequence using the software Blast (version 2.12)[64]. Then, we selected the reads that covered at least 70% of the mapped designed sequence with full identity, which allowed us to obtain the occurrence of each design in the sample. We computed the frequencies $f_{ref}$ of designs in each pool before catalysis, allowing us to quantify the bias in RNA molecules in the initially synthesized pool. Then, we computed the frequencies of designs $f_{sel}$ in the sample with the substrate. To do so, we counted the reads when the substrate was attached right after the 3' end of the design, which indicates that the ribozyme was able to excise the exon and then ligate the substrate. Finally, we computed the experimental activity $act = \log\left(\frac{f_{sel}}{f_{ref}}\right)$. Analyzing reverse reads was sufficient for calculating the activity score. Sequences such that $f_{ref} > 0$ but $f_{sel} = 0$ were considered inactive. Sequences such that $f_{ref} < 5$ were excluded from the analysis.

### Activity threshold and active fraction

As depicted in Fig. 2a, as more mutations are introduced, the activity score decreases from zero (the Azo score reference) to a lower plateau. This plateau is reached for all models beyond 70 mutations, characteristic of non-functional sequences. We considered the distribution of scores at this plateau representative of the experimental noise, taking the score distribution over the 543 sequences comprising at least 100 mutations (red in Supplementary Fig. 4). The noise distribution is found to be Gaussian (Supplementary Fig. 4). Significant activity was taken for a $p$-value $< 10^{-3}$, resulting in a threshold of −2.76. Note that this p-value guarantees a z-score larger than 3.09 (3 standard deviations above the mean). For reference, activities greater than −2.45 corresponded to $p$-value $< 10^{-4}$, and activities greater than −2.16 to $p$-values $\ll 10^{-5}$. Those more stringent thresholds are verified for most of the active sequences found in the $L_{max}$ bin, and the presence of multiple sequences above the threshold further lowers the level of evidence, thus lowering the p-value (Supplementary Table 1). In the case of incubation at different temperatures, we used a more stringent threshold of −1.5 to account for the increased noise level due to shallower sequencing, determined using the same reference pool measured at different sequencing depths. For each model, we computed the active fraction per bin spanning 5 mutations of distance from Azo. For each bin, the active fraction is defined as the number of designs with activity levels above the threshold divided by the total number of designs in that bin.

The error bars on the bins' active fraction are estimated using the error on the individual sequence activity measurements. For this, we used data from 355 overlapping sequences, whose activity has been independently measured three times. As already pointed out, these measurements displayed high consistency, with a correlation coefficient close to 0.99 across different experiments. For each of these 355 sequences, we calculated the standard deviation of the activity measurements. The vast majority of the standard deviations are below 0.3 (with just 5/355 designs exceeding this value). Consequently, we considered 0.3 as the error on the single activity measurement. Using this error estimate, we computed the error bars for the active fraction in each bin by counting the number of designs whose activity values crossed the threshold, considering the ±0.3 measurement error.

### Definitions of $L_{50}$ and $L_{max}$, significance levels

$L_{50}$ was estimated as the largest number of mutations such that a given model achieved at least 50% active sequences within a bin of 5 mutations with $p$-value $< 10^{-3}$ (binomial test, Supplementary Table 1). $L_{max}$ was estimated as the largest number of mutations such that a given model achieved at least 1% success rate within a bin of 5 mutations with $p$-value $< 10^{-3}$ (binomial test, Supplementary Table 1).

However, as the $L_{max}$ bin generally comprises more than 1 sequence, and with scores clearly above the threshold, we computed and reported the p-value given all the available data, rejecting the hypothesis that all sequence activities in this bin are drawn from the noise, as explained below and reported in Supplementary Table 1. To compute this actual $p$-value for $L_{max}$, we assumed a Gaussian null model for the measured activities $a$ of non-functional sequences, $a \sim N(\mu, \sigma)$, which we call the noise model. As described above in the "Activity threshold and active fraction" section, the model mean $\mu$ and standard deviation $\sigma$ are estimated from a sample of 543 non-functional sequences having at least 100 mutations. We assume here that all measured activity is due to experimental noise.

Assume now a sample of $N$ sequences $\{s_i, i = 1, ..., N\}$ with experimentally measured activities $\{a_i, i = 1, ..., N\}$ (typically the sequences of any bin in mutational distance to Azo). For any arbitrary z-value, we can find the number $n(z) = \{i, a_i > \mu + z\sigma\} \vee$ of sequences with measured activity being at least $z$ standard deviations above the mean of the noise model. We can also calculate the probability $\pi(z) = P(a > \mu + z\sigma)$ that a randomly chosen activity $a \sim N(\mu, \sigma)$ from the noise model is beyond that activity threshold.

These two numbers, via a one-sided binomial test, give access to a z-dependent p-value (which allows to refuse or not the hypothesis that all data are drawn from the null model):

$$p = \sum_{i=n(z)}^{N} \frac{N!}{i!(N-i)!} \pi(z)^i [1 - \pi(z)]^{N-i}$$

In coherence with our selection threshold chosen according to the Method section "Activity threshold and active fraction", we report $p$-values for sequences above our standard threshold, i.e., for $\pi(z) = 0.001$, reached at. $z \simeq 3.09$.

In principle, any z-value could be used for calculating p-values. For smaller $z$, we would have more super-threshold activities (larger $n(z)$) but of smaller individual significance (smaller $\pi(z)$), while a larger $z$ would lead to fewer selected sequences of higher individual significance. Note that fixing our activity threshold (and thus considering one value of $z$) leads to an upper bound of the $p$-values.

### Reporting summary

Further information on research design is available in the Nature Portfolio Reporting Summary linked to this article.

## Data availability

Source data for the figures are provided with this paper as a zip folder. Additional data generated in this study have been deposited in the Zenodo database under accession code https://doi.org/10.5281/zenodo.16531362. Source data are provided with this paper.

## Code availability

The code used in this study to generate and analyze sequences has been deposited in the Zenodo database under accession code https://doi.org/10.5281/zenodo.16531362

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

## Acknowledgements
The authors acknowledge Vincent Messow, Nono S. C. Merleau, and Owen Terpstra for preliminary work, Sandeep Ameta, and Michal Matyjasik for discussions. The authors acknowledge the following funding sources: Human Frontier Science Program RGY0077 / 2019 (E.H., M.S., P.N.), Institut Pierre-Gilles de Gennes ANR-10-EQPX-34 (P.N.), -EU H2020 Grant ERC AbioEvo 101002075 (P.N.), EU H2020 grant MSCA-RISE In-ferNet 734439 (M.W.).

## Author contributions
Conceptualization: E.H., M.W., M.S., P.N. Experimental Methodology: C.L., P.P., E.H., P.N. Computational Methodology: V.O., F.C., F.Z., M.W., M.S. Investigation: C.L., V.O., F.C., P.P. Funding acquisition: E.H., M.W., M.S., P.N. Project administration: P.N. Supervision: F.Z., E.H., M.W., M.S., P.N. Writing – original draft: C.L., V.O., P.N. Writing – review & editing: C.L., V.O., F.C., F.Z., E.H., M.W., M.S., P.N.

## Competing interests
Authors declare no competing interests.
