## [Transparent Peer Review file · Nature Communications]

Exploring the space of self-reproducing ribozymes using generative models

Corresponding Author: Dr Philippe Nghe

Version 0:

Reviewer comments:

Reviewer #1

(Remarks to the Author)

In this study, Lambert et al used generative models to produce new RNA sequences, and then experimentally tested these sequences to assess their catalytic activity. Specifically, the authors explore the neutral space of the Azoarcus ribozyme that is capable of reproducing its full-length variant from prefabricated fragments by transesterification, aiming to quantify how many mutated sequences retain catalytic activity. By generating and testing sequences with increasing mutations, they assess how far the neutral space extends from the original sequence. The generative models used to generate the random RNAs are varied in complexity (e.g., Direct Coupling Analysis, random mutagenesis) and were designed to explore how well they could introduce mutations while retaining functionality. The size of the neutral space is then estimated using statistical models and experimental data. Technically and experimentally, this is an interesting study with significant contributions to understanding RNA functionality. However, several critical conceptual issues, particularly regarding the interpretation of evolutionary processes and prebiotic plausibility, must be addressed.

Major points:

1. The current manuscript is concise, but the relaxed length requirements by Nature Communications provide an opportunity to expand the introduction. For instance, the term 'neutral network' might confuse readers unfamiliar with it. Including a clear definition and providing more context about its prebiotic significance would improve accessibility.
2. The focus on the Azoarcus ribozyme, while appropriate as a model system, limits the generalizability of the findings. Other ribozymes, in particular those with complex functions (e.g. polymerization) likely have stricter constraints, reducing their neutral space. Explicitly addressing whether these results apply to other ribozymes would strengthen the manuscript.
3. The estimate ($\sim 10^{39}$) is derived from models such as DCA, which rely on assumptions about sequence conservation and covariation that are rooted in modern evolutionary constraints. These models do not necessarily reflect the constraints of prebiotic environments. Explicitly comparing how the DCA predictions differ from what might be expected in prebiotic environments would strengthen the discussion.
4. The study suggests that their findings support a diverse RNA world and alternative paths for abiogenesis, but this assumes the generalizability of the neutral space concept across different functional ribozymes. A more rigorous argument is needed to connect these findings to prebiotic plausibility.
5. For this work to serve as evidence for abiogenesis, it needs to address whether these sequences could plausibly arise and sustain function in early Earth conditions. The bold sentence "The density of self-reproducers may thus be many decades higher, making evidence for a probable RNA origin of life within reach." is very misleading in my opinion and should be toned down.
6. The calculation in the SI gives the impression that nature blindly samples RNA sequences until a self-reproducing 197-mer emerges. This misrepresents how prebiotic evolution likely occurred. Instead, abiogenesis is better understood as a

stepwise, incremental process involving chemical biases, selective enrichment, and cumulative improvements. The emergence of long RNAs capable of autocatalytic self-reproduction likely followed a stepwise pathway: Early, shorter oligomers may have exhibited primitive catalytic or structural functions. These sequences might have been augmented through processes like non-enzymatic polymerization, ligation, or fragment recombination. Jumping directly to a 197-nucleotide ribozyme ignores this gradual emergence of functional complexity. While the authors' frequency estimate is valuable, it should be presented within the context of these gradual processes to avoid oversimplifying the complex mechanisms underlying the origin of life.

7. Have the authors considered retraining their models using experimentally validated sequences? This could improve the predictive power of the models and expand their exploration of neutral space.

Minor points:

1. The abstract could be revised for clarity and precision, avoiding redundancy (e.g., repeated mentions of 'RNA self-reproduction') and providing a clearer explanation of key terms like 'neutral space.' Additionally, the phrase 'abiogenesis could have followed many alternative paths' would benefit from further elaboration to contextualise the findings.
2. The phrase 'interpolates with Azo' is vague. Providing a more precise explanation, such as 'DCA-generated sequences bridge the sequence space between Azo and natural chimeras,' would clarify its meaning.

(Remarks on code availability)

Reviewer #2

(Remarks to the Author)

Summary

The manuscript describes a computational exploration of the neutral sequence space of a trans-splicing Azoarcus ribozyme. The computational method employs generative probabilistic models based on statistical learning and structure prediction, combined with a high-throughput assay of 24,220 unique RNA sequences for catalytic activity. The study finds that randomization while maintaining canonical basepairs and introducing randomization in loops while leaving tertiary contacts intact increased the success rate of finding active ribozyme variants. This is not new because comparative sequence analysis of group I introns, and biochemical analysis of group I intron variants has previously found the same (Michel & Westhof 1990, J.Mol.Biol. 216, 585), and it should be pointed out as such. The new aspect - how statistical learning can improve the predictions and increase the estimates of catalytically active variants in sequence space - is impressive but also very hard to understand. The current manuscript fails to describe clearly what the new method does (in contrast, it describes clear and concisely what the secondary structure based constraints do). As figure 4 shows, the method seems to work well because the results are validated by wet-lab experiments, which convinces me that the study would be impactful enough for Nature Communications. However, the presentation (in text and figures) as well as the interpretation (with regard to origins of life) needs significant effort to make the study accessible enough, and rigorous enough for Nature Communications.

major points

1 - Fig.1 is central to understanding the manuscript and would benefit from a few changes. In the current form, it took me 30 minutes to understand the figure although I know this ribozyme well.

Fig 1a: I appreciate that every nucleotide is shown explicitly in the context of the secondary structure. Please explicitly annotate the 5'- and 3'-termini with 5' and 3'. Please make clear that the 5'-terminus is not covalently connected to the phosphodiester bond between nucleotides 122 and 123, and that the omega-G is not covalently connected to the bond between positions 118 and 119. This could be done by moving the 5'-terminus and P1 duplex below the two crossing bonds (122-123 and 38-39), and placing the bond between nucleotides 118 and 119 to the right of the omega-G so that there is no crossing. It may also help clarity if the domain with positions 39-117 are placed to the right as shown in figure 1B of Dolan & Muller 2014, RNA 20, 203.

Fig. 1c: Please show the ribozyme in the same format as in figure 1a. In the current format, only the reader with prior knowledge of group I intron ribozymes knows that the circled G refers to the gray G in figure 1a, the three base pairs at the other ribozyme end refer to the three gray nucleotides, and the orientation of the two substrates has the G at the 5'-terminus and the hydroxyl at the 3'-terminus. The size of the secondary structure in figure 1a fits well to replace each of the four steps shown in figure 1c.

Fig. 1d: Please remove the gray egg-shaped illustration; it is not necessary and was distracting / confusing to me.

2 - Several paragraphs with overly technical language made it impossible for me to understand the core achievements of the manuscripts in a reasonable time.

The paragraph starting at line 111 is hard to understand. The first sentence reveals only after reading multiple times that it defines omegaDCA; please place the character omegaDCA in parentheses after the definition, not before. In line 113, the sentence "Indeed, DCA biases predictions towards designs to which it attributes 113 high probability. " does not seem to make sense. Of course, a software designed to find high model success will bias towards high model success. Please clarify. The next sentence cites "Standard results 115 in information theory²⁴ " without mentioning what these are. Please add a half-sentence that makes clear to the reader what approach was used, or what data / theorems were processed to obtain these standard results. Even after reading multiple times I was not able to understand the next sentences ("Ω can be

interpreted as the effective number of different sequences that the model can generate (the so-called support size of the model) [Please clarify what limits the model from generating more sequences]. Intuitively, sampling a finite number of sequences from the probability distribution $P(x)$ is highly likely to be part of this subset of Ω sequences out of the 4L possible sequences [isn't that the definition of what the model should do?]. Although it is not known how to compute Ω in general, we devised a semi-analytical method exploiting the structure of DCA."). While the results sound impressive ("The success rate at $L_{max}=60$ mutations was determined to be larger than 1% with $p < 1264 \cdot 10^{-5}$ confidence.") I don't understand the procedure that was described in this paragraph.

The paragraph starting in line 128 seems to describe that the most important factor for activity is the closeness to the sequence of the Azoarcus ribozyme (second sentence). The next sentence ("To quantify the variety of sequences generated relative to the 132 natural diversity, ") took me a while to understand - please make it simpler. At the end of this paragraph, please clarify also which results are directly supported by wet-lab analysis, and which are computational results.

3 - Lines 219-233: The results are over-interpreted with respect to origins of life (for example line 233: "...making evidence for a probable RNA origin of life within reach."). First, the self-reproducing ligase ribozymes and Azoarcus ribozymes discussed in this study can only reproduce themselves from RNA fragments with very specific sequences. In a prebiotic environment it is hard to envision how such fragments could have existed, and in the absence of such fragments, no replication results. Second, self-replication does not equal life. Without open-ended evolution, the replicating system, unable to invent new catalytic activities, eventually dies out. Additionally, the results found with the Azoarcus ribozyme are non-quantitative, and the current success of the model is based on a few instances among the 24,000 tested RNA sequences.

minor points

abstract, second-last line: typo: it should be "... can be addressed ..."

line 40: it should be '.. a neutral network...' or 'the corresponding neutral network' but not 'the neutral network because it has not yet been specified in the introduction what neutral network it refers to.

line 78: Is it a typo that the acronym RDM is used for 'random uniform mutagenesis'? Please make clear why RDM is the right acronym, or use RUM instead.

The manuscript contains several three-letter acronyms that are unnecessary and make reading hard (RDM, DCA, MSA, CHI, BPR, VAE). Please minimize the use of acronyms. For example, there is no reason to use MSA; this acronym is used only once more in addition to its original use and definition, and it is much easier for the reader if the term is spelled out twice. CHI also seems unnecessary. While 'Azo' is often used by researchers in the field, please avoid this lab jargon and instead use the full name 'Azoarcus ribozyme'.

(Remarks on code availability)

Version 1:

Reviewer comments:

Reviewer #1

(Remarks to the Author)

The authors have addressed almost all my points satisfactorily. However, I remain unconvinced of the necessity and usefulness of the probability calculation as it is currently presented in the discussion. I agree with the new (and well-justified) statements that this number should not be taken literally, since it should not be interpreted as a meaningful measure of abiogenic plausibility. But what conceptual purpose does it serve then? In fact, this kind of framing is prone to misinterpretation, especially in public or interdisciplinary discourse, where such probabilistic arguments have been (mis)used in the past, e.g. to support intelligent design narratives.

Despite the subsequent qualifications, the limitations appear almost as afterthoughts, logically subordinate to a statistical framework that itself rests on highly constrained and arguably artificial assumptions. In my view, by including this calculation, the authors risk reinforcing the very misconception they are trying to disarm: that the origin of life depends on the improbable spontaneous appearance of a highly complex, functional replicator.

I therefore strongly recommend reorienting the discussion to focus on the compelling empirical and modeling results regarding ribozyme diversity and either remove the probability estimate altogether and discussing things more qualitatively, or adding or combining it with probability estimates of much shorter replicators (which have been found experimentally, as the authors' themselves point out) by extrapolating their findings from the azoarcus-type ribozymes to simpler ribozymes. This would allow the manuscript's real contributions (namely, the generative modeling and exploration of neutral spaces using high-throughput methods) to occupy the conceptual center they deserve, while avoiding the risk of unintentionally perpetuating outdated or misleading views of abiogenesis.

(Remarks on code availability)

Reviewer #3

(Remarks to the Author)

In this study, Lambert et al. tested different models for generating active ribozyme RNA sequences from a reference self-reproducing ribozyme. The active sequences were identified using a high-throughput assay coupled with deep sequencing. The combination of the evolutionary model DCA with a structure-based score (SB) showed the best performance in generating long-distance mutations that retained ribozyme activity. The neutral space was computed after correction using experimental data. This study is experimentally sound; however, beyond providing a lower-bound estimate for the number of active ribozymes (as better models might yield even higher numbers), the new insights this work contributes to the ribozyme or origin-of-life research community should be described more clearly.

Other detailed Comments:

1. The title “Discovering self-reproducing ribozymes...” is broad and could imply, for example, that the presented models generate ribozyme variants with improved catalytic activity or kinetics over the reference sequence, or that they create active ribozymes from partially fixed folding scaffolds. However, this work demonstrates that the models generate sequences with long-distance mutations that retain partial activity of the reference ribozyme from an MSA input. Other properties such as folding robustness or thermostability—important characteristics in prebiotic evolutionary processes—were not analyzed or discussed. In my opinion, the title should be revised to better reflect the actual content of the research, such as: “Exploring the Activity Neutral Network of Self-Reproducing Ribozymes...”

2. The correlation between the gel shift assay–verified activity and the activity score or model-generated score (if available) should be presented and discussed.

3. In Figures 2a–b and 5b, only the active fraction is shown, which omits the information on the activity levels of the variants. The distribution of activity scores, or the mean/median activity score (as shown in Fig. S1), should be included in the main text to provide a more comprehensive view of catalytic activity.

4. A discussion on measuring the activity of the designed sequences at different (especially higher) temperatures, or performing the high-throughput assay under varied temperature conditions, could be added. This might offer additional insights into the structure of the neutral set and potential evolutionary trajectories.

(Remarks on code availability)

Version 2:

Reviewer comments:

Reviewer #3

(Remarks to the Author)

The authors have addressed my concerns and improved the quality of the manuscript with extra results and discussion. I recommend publication of this manuscript in Nature Communications.

(Remarks on code availability)

Reviewer #1 (Remarks to the Author):

In this study, Lambert et al used generative models to produce new RNA sequences, and then experimentally tested these sequences to assess their catalytic activity. Specifically, the authors explore the neutral space of the Azoarcus ribozyme that is capable of reproducing its full-length variant from prefabricated fragments by transesterification, aiming to quantify how many mutated sequences retain catalytic activity. By generating and testing sequences with increasing mutations, they assess how far the neutral space extends from the original sequence. The generative models used to generate the random RNAs are varied in complexity (e.g., Direct Coupling Analysis, random mutagenesis) and were designed to explore how well they could introduce mutations while retaining functionality. The size of the neutral space is then estimated using statistical models and experimental data. Technically and experimentally, this is an interesting study with significant contributions to understanding RNA functionality. However, several critical conceptual issues, particularly regarding the interpretation of evolutionary processes and prebiotic plausibility, must be addressed.

We thank the reviewer for her or his positive and helpful comments, which we address below. We have notably expanded the introduction and discussion, as well as added a novel section with additional results presented in a new Figure 5 and a new Supplementary Figure 11.

Major points:

1. The current manuscript is concise, but the relaxed length requirements by Nature Communications provide an opportunity to expand the introduction. For instance, the term 'neutral network' might confuse readers unfamiliar with it. Including a clear definition and providing more context about its prebiotic significance would improve accessibility.

We have reworked a longer introduction, specifically to introduce the terms 'neutral network' or 'neutral set' in the second paragraph:

“The RNA world hypothesis surmises that early evolving systems consisted of reactions between RNAs catalyzed by RNAs¹. In this scenario, the first ribozymes (catalytic RNAs) could have arisen by chance from random polymerization, possibly helped by covalent assembly reactions between partially complementary RNAs^{2,3}. Among those, certain ribozymes, or sets of ribozymes, would amplify by catalyzing their own production, a process called autocatalysis^{4,5}. Autocatalytic amplification of ribozymes from shorter, preformed oligonucleotides has been demonstrated experimentally in two RNA systems: reciprocal ligases⁶ and recombinases derived from self-splicing RNAs⁷. It is envisioned that a gradual process of evolution could have lead from autocatalytic RNAs⁸ to template-based replication of RNA catalyzed by polymerase ribozymes^{9,10}, thereby initiating a mode of evolution similar to biological evolution.

The plausibility of the RNA world thus relies on the hypothesis that RNA self-reproduction is sufficiently common among all possible sequences of RNA to have emerged by chance. However, due to the scarcity of known self-reproducing RNA systems, there is a lack of evidence to support this hypothesis. This requires to characterize the size of the *neutral network*, or *neutral set*, defined as all sequences carrying out the same activity¹¹, namely RNA self-reproduction. A general strategy to explore this neutral set is to

diversify RNAs known to have the desired property, and assess how many of the resulting sequences conserve this property. It would then be possible to estimate the frequency of autocatalytic RNA in sequence space.”

2. The focus on the *Azoarcus* ribozyme, while appropriate as a model system, limits the generalizability of the findings. Other ribozymes, in particular those with complex functions (e.g. polymerization) likely have stricter constraints, reducing their neutral space. Explicitly addressing whether these results apply to other ribozymes would strengthen the manuscript.

Regarding generalization of the findings, either there exist a sufficient number of analog sequences to learn from (typically a naturally occurring family of ribozymes), or the sequences were found by artificial evolution so only one (or a few) is known. Unfortunately, replicase ribozymes are in the latter category and it is not possible to apply our strategy. However, by examining a large number of natural families, we found general trends relating diversity and length, which can be used as a first guess to extrapolate our support size estimations in general. To address the point of the reviewer to be best we could, given the available data, we performed two additional analyses:

- We highlight the diversification of another ribozyme from the same group I intron family which we comment on in a new paragraph of a new result section (formerly, it was only briefly mentioned in the conclusion). In this case, we could perform the generative DCA and corresponding experiment. Although incremental, we think this result is a first doable step to show generality.
- To extrapolate the findings, we expanded a formerly published result on the scaling laws of DCA support size as a function of length (Calvanese et al. *Nucleic Acid Research* 2024). This work shows that theoretical support size varies as a power law of the length across many RNA families analyzed in the range 50-300 nucleotides. To more specifically address the case of ribozymes, we computed the support size of 3 other ribozyme families, considering different lengths as way to compare from simpler to more complex ones relative to *Azoarcus*: a shorter one (Hammerhead), a similar length one (the glmS riboswitch ribozyme), and a longer one (the bacterial RNase P class A). The analysis is presented in a new Supplementary Figure 11 and confirms the trends of diversity as a function of the number of mutations found for *Azoarcus* for these 3 families, as well as agreement with the general power law of diversity as a function of length. A dedicated section has been added in the new result section and the discussion.

Additional paragraph in the new result section:

“Finally, we wondered if the results were specific to the *Azoarcus* ribozyme model system, or if it could be extrapolated to other ribozymes. First, we applied the DCA model using a group I intron from *Phormidium* sp. as a starting point for mutagenesis (Supplementary Fig. 10, Supplementary Information ‘Phormidium’). We measured $L_{\max}=25$ for the DCA model and $L_{\max}=35$ for the DCA-SB model. Although the degree of diversification was lower than for *Azoarcus*, this indicates that the diversification approach can be generalized to extend the neutral set from other starting points. A next question is whether these findings might be applied to ribozymes that differ by their

activity, structure, or length. A recent analysis shows that the effective number of RNAs with a given activity within a family of structural analogs can be estimated, according to DCA, using a power law d^L , where L is the length of the RNA and d is an effective nucleotide diversity per position found to be 1.74³². Following this analysis, we more computationally examined three ribozyme families with diverse lengths: Hammerhead (66 nucleotides), glmS (167 nucleotides), and Bacterial RNase P class A (367 nucleotides). We observed comparable trend of the support size versus number of mutations, which furthermore matched the scaling law of support size versus length formerly found on a large number of naturally-occurring RNAs³² (Supplementary Fig. 11). These results confirm the generality of generative approaches for functional RNAs, also found by a deep learning method²⁶, and indicates a rule of thumb to extrapolate our DCA-based estimation of diversity to other ribozymes.”

New Supplementary Figure 11:

Supplementary Figure 11. Diversification potential of ribozymes. a) Curve of accessible diversity (support size) as a function of the number of mutations introduced in an initial sequence of the RF02276 RNA family of Hammerhead ribozymes (458 members, 66 nucleotides) for Random Uniform Mutagenesis, the Profile model (position-wise nucleotide frequencies), and the DCA model. Each row represents the frequency of mutations in all four nucleotides where dots represent the wild type nucleotides. b) Same as panel (a) for the RF00234 RNA family of glmS riboswitch ribozymes (943 members, 167 nucleotides). c) Same as panel (a) for the RF00010 RNA family of Bacterial RNase P class A (8929 members, average length 367 nucleotides). d) Scaling laws of diversity as a function of length, where the DCA support size estimate is shown for the families of panels (a-c).

3. The estimate ($\sim 10^{39}$) is derived from models such as DCA, which rely on assumptions about sequence conservation and covariation that are rooted in modern evolutionary constraints. These models do not necessarily reflect the constraints of prebiotic environments. Explicitly comparing how the DCA predictions differ from what might be expected in prebiotic environments would strengthen the discussion.

This is an interesting point that led us to test a lower magnesium concentration of 5 mM instead of 60 mM, a critical parameter that may vary between environmental scenarios. We have a panel b in the new Figure 5 and a corresponding paragraph in the new final section of the results:

“To relate our findings to possible prebiotic scenarios, we assessed the impact of magnesium concentration, a critical parameter known to affect structural stability and catalysis in RNA. While the experiments reported so far were done at 60 mM magnesium concentration, we used a 5 mM magnesium concentration to run the assay for the pools of the RUM and DCA-SB, the models that introduced the least and the most mutations, respectively. As expected, the active fraction decreased in both cases, consistently in each mutation bin (Fig. 5b). The impact on the RUM was dramatic, with $L_{50}=0$ and $L_{max}=5$, due to initially low metrics. However, the maximum diversification was found to be robust for the DCA-SB model, with $L_{50}=30$ and $L_{max}=60$.”

Fig. 5. Robustness and generality. a, Designed DCA sequences at 60 mutations from the WT. The histogram shows the activity scores of 991 detected designs (out of 1000 designs) with 22 active sequences beyond the activity threshold (equal to -2.76, red dotted line), corresponding to 2% of the pool. b, Comparison of active fraction as a function of the number of mutations for the RUM and DCA-SB models, for different MgCl₂ concentrations: 60mM as our standard condition (solid lines) and a lower concentration of 5 mM (dashed lines). Dots are the average activity per bin of 5 mutations and bars represent the 25-75 percentile range.

4. The study suggests that their findings support a diverse RNA world and alternative paths for abiogenesis, but this assumes the generalizability of the neutral space concept across different functional ribozymes. A more rigorous argument is needed to connect these findings to prebiotic plausibility.

We agree and based on the results added to answer point 2 and existing literature, we have added a paragraph in the discussion about generalization to other ribozymes and their possible role during abiogenesis:

“Further, it is likely that smaller and simpler instances of autocatalytic RNAs contributed to prebiotic evolution. The Azoarcus ribozyme is a model for proof-of-concept studies of RNA chemical reactions in the RNA world. Smaller group I intron analogs are known to exist, such as a 140 nucleotides long ribozyme derived from a shortened SunY ribozyme. Engineered autocatalytic ligases are 67 nucleotides long, overall relying on a complex of 134 nucleotides⁶, although these require activated oligomers. Even smaller ribozymes were recently found to emerge from random mixtures of activated oligomers, with lengths as short as 20 nucleotides⁴⁴, relying on 40 nucleotides complexes. Scaling laws relating ribozyme length and diversity, as shown above, could allow us to estimate the size of their neutral sets³².”

Note that the text additions to answer to point 6 also contribute to the generalization.

5. For this work to serve as evidence for abiogenesis, it needs to address whether these sequences could plausibly arise and sustain function in early Earth conditions. The bold sentence "The density of self-reproducers may thus be many decades higher, making evidence for a probable RNA origin of life within reach." is very misleading in my opinion and should be toned down.

Our comment was indeed too speculative. We have toned down the last sentence in the abstract: “Our findings show how the origin of life question can be addressed quantitatively and statistically rather than on the basis of singular instances, suggesting that abiogenesis could have followed a diversity of paths.”

We also reformulated related claims in a more nuanced manner in the discussion:

“Considering a scenario where autocatalytic RNAs may emerge from random mixtures of smaller oligomers, we computed that a concentration of RNA in the molar range would be required for a self-reproducer to be found somewhere in the observable universe over a billion year (Supplementary Information ‘Plausibility of RNA self-reproduction’). Obviously, this estimation does not prove that an RNA emergence of life is probable. Nevertheless, it sets the problem within the scales of physics and chemistry.

Importantly, this estimation should not be taken literally. First, it is only a lower bound for generated from one member of one family of Group I introns. We also showed that diversification

could be performed from different starting ribozymes (Azoarcus and Phormodium ribozymes). On the longer term, generating RNAs with the same activity but much less sequence identity might be possible as it is possible in proteins⁴³. This estimate is also based on the detection limit of our experiments, and allowing for lower activity could have a significant effect on this number. Longer reaction times and deeper sequencing would be expected to expand the neutral set. Beyond group I intron analogs, relaxing the structural constraints would further expand the range of RNAs capable of self-reproducing. For instance, the family of group II introns is structurally distinct and also catalyzes recombination reactions between RNAs.

[...] These studies suggest that ribozymes are, in general, possibly many orders of magnitude more frequent than we estimated.

Ultimately, generative approaches with machine learning, as developed in this work, could uncover the full extent of RNA sequences capable to achieve functions relevant to the RNA world. Such systematic molecular diversification may ultimately reveal the variety of accessible pathways for prebiotic evolution.”

Reformulation of the corresponding section in the Supplementary Information:

“Future estimations may lead to improve this bound by several decades, as indicated by estimations based on the size of libraries used in directed evolution. This rough calculation is only intended to give an idea of our state of knowledge, and any true assessment would require to consider a multistep scenario for abiogenesis, as discussed in the main text.”

6. The calculation in the SI gives the impression that nature blindly samples RNA sequences until a self-reproducing 197-mer emerges. This misrepresents how prebiotic evolution likely occurred. Instead, abiogenesis is better understood as a stepwise, incremental process involving chemical biases, selective enrichment, and cumulative improvements. The emergence of long RNAs capable of autocatalytic self-reproduction likely followed a stepwise pathway: Early, shorter oligomers may have exhibited primitive catalytic or structural functions. These sequences might have been augmented through processes like non-enzymatic polymerization, ligation, or fragment recombination. Jumping directly to a 197-nucleotide ribozyme ignores this gradual emergence of functional complexity. While the authors’ frequency estimate is valuable, it should be presented within the context of these gradual processes to avoid oversimplifying the complex mechanisms underlying the origin of life.

We fully agree that prebiotic evolution is likely to be a multistep gradual process. We dedicated a new paragraph to this point:

“Finally, autocatalytic RNAs must be considered in the larger picture of abiogenesis. Certain prebiotic evolutionary processes could have facilitated their appearance. Pre-assemblies of oligomers with partial complementarity may have enriched for ribozyme building blocks^{2,3}. Interactions with the geological milieu can also favor structured RNAs that are prone to catalysis⁴⁵. However, the prebiotic milieu could be constraining. In particular, lower magnesium concentration tends to reduce the fraction of active molecules, which we however found to have a limited impact on our estimation of diversity in the mM range. Early evolutionary processes in the RNA world

probably required cooperation between diverse molecular species (e.g. peptides and lipids) and catalytic activities. Although this question has been scarcely addressed to date, the frequency of other important activities has been probed by randomization techniques. Ligase ribozymes, from which replicases were derived, were selected from random pools of size 10^{1546} . The diversification potential of aminoacylation ribozymes, relevant to origin of translation, was explored by mutagenesis in a subregion of length 21 nucleotides, showing that 10^5 sequences were active among the $4 \cdot 10^{12}$ possible ones²².”

7. Have the authors considered retraining their models using experimentally validated sequences? This could improve the predictive power of the models and expand their exploration of neutral space.

Retraining from our data is an exciting possibility, which we in fact tried. The result of the retrained DCA called RTDCA can be seen as the dark blue curve analog of our Figure 2, below. Although the success rate is slightly improved around 20 mutations, RTDCA was less predictive at larger distances. We believe this to be due to a naive implementation of the retraining, for which used the set of successful sequences with uniform weighting. This approach appears to be poor at using information from the large mutation range because this region is the least populated among successful sequences. Relearning efficiently from generated data is an active area of research in machine learning, and it is in particular an open challenge to integrate negative data as well as correcting for sampling biases. This is the focus of a contribution in the near future. We have decided to show the data for the purpose of the present review but not in the article, as we felt that it would be confusing to show and explain a partial result.

Minor points:

1. The abstract could be revised for clarity and precision, avoiding redundancy (e.g., repeated mentions of ‘RNA self-reproduction’) and providing a clearer explanation of key terms like ‘neutral space.’ Additionally, the phrase ‘abiogenesis could have followed many alternative paths’ would benefit from further elaboration to contextualise the findings.

We do not mention the notion of ‘neutral set’ in the abstract due to stronger length restrictions in Nature Communications as compared to the original submission, but we now define it early in the introduction. We have simplified the abstract, starting with:

“Estimating the plausibility of self-reproduction is central to origin-of-life scenarios. However, this property has been shown in only a handful of catalytic RNAs. Here, we show the existence of a very large number of RNAs that assemble themselves from oligomers, covering a vast sequence space.”

We end the abstract using a lighter tone, knowing that the discussion has been expanded on this:

“Our findings show how the origin of life question can be addressed quantitatively and statistically rather than on the basis of singular instances, suggesting that abiogenesis could have followed a diversity of paths.”

2. The phrase ‘interpolates with Azo’ is vague. Providing a more precise explanation, such as ‘DCA-generated sequences bridge the sequence space between Azo and natural chimeras,’ would clarify its meaning.

We have cut the sentence into two and used a slightly modified version of the suggestion, to specify that interpolation is understood as number of mutations introduced:

“We found that the chimeras (representative of natural diversity) clustered near the Azoarcus ribozyme in these projections (Fig. 2c), while the DCA-generated sequences bridge the sequence space between the Azoarcus ribozyme and natural chimeras, from the perspective of the first two principal components. We note that PC1 strongly correlates with the distance to the Azoarcus ribozyme (Pearson $\rho=0.88$, $p<10^{-5}$).”

Reviewer #2 (Remarks to the Author):

Summary

The manuscript describes a computational exploration of the neutral sequence space of a trans-splicing Azoarcus ribozyme. The computational method employs generative probabilistic models based on statistical learning and structure prediction, combined with a high-throughput assay of 24,220 unique RNA sequences for catalytic activity. The study finds that randomization while maintaining canonical basepairs and introducing randomization in loops while leaving tertiary contacts intact increased the success rate of finding active ribozyme variants. This is not new

because comparative sequence analysis of group I introns, and biochemical analysis of group I intron variants has previously found the same (Michel & Westhof 1990, J.Mol.Biol. 216, 585), and it should be pointed out as such. The new aspect - how statistical learning can improve the predictions and increase the estimates of catalytically active variants in sequence space - is impressive but also very hard to understand. The current manuscript fails to describe clearly what the new method does (in contrast, it describes clear and concisely what the secondary structure based constraints do). As figure 4 shows, the method seems to work well because the results are validated by wet-lab experiments, which convinces me that the study would be impactful enough for Nature Communications. However, the presentation (in text and figures) as well as the interpretation (with regard to origins of life) needs significant effort to make the study accessible enough, and rigorous enough for Nature Communications.

We thank the reviewer for his or her assessment and remarks. We address the different points below. We have added a sentence in the last paragraph of our introduction to motivate our approach, acknowledging the contribution of Michel & Westhof 1990, J.Mol.Biol. 216, 585:

“However, none of these studies have characterized the size of the neutral set for autocatalytic RNA self-reproduction. Analysis of evolutionary conservation and structure in naturally occurring group I introns (Fig. 1a) suggests a large potential neutral set that has not yet been experimentally evaluated²⁷.”

major points

1 - Fig.1 is central to understanding the manuscript and would benefit from a few changes. In the current form, it took me 30 minutes to understand the figure although I know this ribozyme well. Fig 1a: I appreciate that every nucleotide is shown explicitly in the context of the secondary structure. Please explicitly annotate the 5'- and 3'-termini with 5' and 3'. Please make clear that the 5'-terminus is not covalently connected to the phosphodiester bond between nucleotides 122 and 123, and that the omega-G is not covalently connected to the bond between positions 118 and 119. This could be done by moving the 5'-terminus and P1 duplex below the two crossing bonds (122-123 and 38-39), and placing the bond between nucleotides 118 and 119 to the right of the omega-G so that there is no crossing. It may also help clarity if the domain with positions 39-117 are placed to the right as shown in figure 1B of Dolan & Muller 2014, RNA 20, 203.

We have applied all except one of the requested changes in Figure 1 (see below for the modified Figure 1). In particular, it was not clear where the 5' and 3' extremities were, and we now indicate them as recommended in Figure 1 as well as other appearances of the ribozyme in Figures 4d and Supplementary Figure 7a. However, we did not follow the suggestion of putting the domain with positions 39-117 to the right of the molecule. We understand that this could have the advantage to limit the entanglement in the representation of the structure. However, crossings cannot be fully avoided and having the scaffold region on the left seems to be the most customary choice in the literature. Although this choice may result from some historical contingency, we preferred to keep to the most common representation.

Fig. 1c: Please show the ribozyme in the same format as in figure 1a. In the current format, only the reader with prior knowledge of group I intron ribozymes knows that the circled G refers to the gray G in figure 1a, the three base pairs at the other ribozyme end refer to the three gray nucleotides, and the orientation of the two substrates has the G at the 5'-terminus and the hydroxyl at the 3'-terminus. The size of the secondary structure in figure 1a fits well to replace each of the four steps shown in figure 1c.

Done. However, note that the changes above that clarify the representation of the structure in panel a could not be easily applied to the 5' extremity in this context because the substrate would then cross with other parts of the molecule. We thought to put the 5' extremity higher in this case. We hope the representation remains clear enough.

Fig. 1d: Please remove the gray egg-shaped illustration; it is not necessary and was distracting / confusing to me.

Done.

2 - Several paragraphs with overly technical language made it impossible for me to understand the core achievements of the manuscripts in a reasonable time.

We agree that the descriptions of support sizes were unclear, in part because they were far too short. We have fully reworked them, now appearing in red in the section "Populating the neutral space with Direct Coupling Analysis". We have also added explanations about the underlying results from information theory in a dedicated section of the Supplementary Information (see below).

The paragraph starting at line 111 is hard to understand. The first sentence reveals only after reading multiple times that it defines Ω_{DCA} ; please place the character Ω_{DCA} in parentheses after the definition, not before.

We now start as follows:

"We next set out to use this DCA model to estimate the size of the neutral set of this type of autocatalytic self-reproducing RNA. It is important to note that the number of potential autocatalytic self-reproducers Ω_{DCA} predicted from the DCA model cannot be simply calculated by multiplying the total number of possible sequences at mutational distance L by the measured active fraction generated by DCA."

In line 113, the sentence "Indeed, DCA biases predictions towards designs to which it attributes 113 high probability. " does not seem to make sense. Of course, a software designed to find high model success will bias towards high model success. Please clarify.

The reviewer is right, the sentence was not informative. It makes sense only when contrasted to random uniform mutagenesis. We now say:

"Indeed, the DCA model generates predictions focused on a restricted subset of sequences x as compared to random uniform mutagenesis, namely the sequences to which DCA assigns a high probability $P(x)$. Thus, the number of sequences among which the DCA model samples is in practice smaller than the total number of possible sequences."

The next sentence cites "Standard results 115 in information theory²⁴ " without mentioning what these are. Please add a half-sentence that makes clear to the reader what approach was used, or what data / theorems were processed to obtain these standard results. Even after reading multiple times I was not able to understand the next sentences (" Ω can be interpreted as the effective number of different sequences that the model can generate (the so-called support size of the model) [Please clarify what limits the model from generating more sequences]. Intuitively, sampling a finite number of sequences from the probability distribution $\square(\square)$ is highly likely to be part of this subset of Ω sequences out of the 4^L possible sequences [isn't that the definition of what the model should do?]. [...]

We acknowledge that the initial manuscript was far too elusive on this key result. The underlying theory is provided in Chapter 3 of the book of Cover and Thomas "Elements of Information Theory". It justifies from the viewpoint of probability theory the fundamental principle of statistical

physics, namely that entropy is the logarithm of the number of effective states of the system. In principle, nothing limits the model from generating more sequences, as we now mention (“On the other hand, to restrict the size of space given the distribution P is not straightforward because any sequence may have a non-zero, however small, probability.”). However, when sampling in practice, some samples are so rare compared to others that it becomes exponentially less likely to draw them as compared to resample an already seen one. This result is made precise in Cover and Thomas, which we now summarize in the main text and Supplementary Information.

To avoid crowding the main text, we report only the final result useful for our purpose:

“In information theory, the “effective support size” is a term used to define the effective size of a sampling set³⁰. The effective support size can be defined for any probability distribution P over all sequences x as $\Omega = \exp(S)$, where $S = -\sum P(x) \log P(x)$ is the Shannon entropy, the sum being taken over all sequences x . This definition relies on the following mathematical result: when sampling N sequences x from the probability distribution P , the sample probability behaves like $\left(\frac{1}{\Omega}\right)^N$ for large enough N (see Supplementary Information for more technical details^{30,31}). Intuitively, this means that sampling from the model follows the same statistics as if one were sampling uniformly from a subset of size Ω . In other words, the vast majority of the probability is concentrated in a subset of size Ω , interpreted as the effective number of different sequences that the model can generate.”

We added a detailed section in the Supplementary Information a more technical summary of the Cover and Thomas results for the interested reader:

“Support size computations

In a strict sense, the support X of a probability distribution $P(x)$ is the set of all outcomes x that have a non-zero probability, and the support size is the cardinal Ω of X . The *effective* support size is an estimation of the number of different outcomes you can expect in practice when sampling from the distribution. For instance, with a fair dice, you expect 6 different outcomes. However, for a rigged dice where the face displaying ‘6’ has a probability of 1/1000 instead of 1/6, you would expect an effective support size closer to 5 because this face is rarely seen. One measure of the effective support is the exponential of the entropy of the probability distribution $S = -\sum_{x \in X} P(x) \log P(x)$. In the dice example, the effective support size of the fair dice computed from entropy equals $2^{2.58\dots} = 6$ and equals the actual support size. For the rigged dice the effective support size is $2^{2.33\dots} = 5.03$, consistent with our intuitive expectation. This choice is justified from results of the book of Cover and Thomas *Elements of Information Theory* cited in the main text, summarized below.

For models such as Random Uniform Mutagenesis, Random Base Pairs, Random Base Pairs with 3D constraints, the distribution $P(x) = \frac{1}{\Omega}$ is equiprobable over Ω states, and

$$S = -\sum_{x \in X} P(x) \log P(x) = \sum_{x \in X} \frac{1}{\Omega} \log \frac{1}{\Omega} = \log \Omega.$$

As in the fair dice example, the actual support size Ω of uniform probability distributions coincides with the effective support size computed from the entropy as $\exp(S) = \Omega$.

In the non-equiprobable cases (Profile, SB, DCA, DCA-SB), consider N independent samples $x^i \in X$ from a distribution $P(x)$ with entropy S . For any $\varepsilon > 0$, one defines the *typical set* $A_\varepsilon^{(N)} \subset X^N$ as the set of N -tuples verifying:

$$e^{-N(S+\varepsilon)} \leq P(x^1, x^2, \dots, x^N) \leq e^{-N(S-\varepsilon)}.$$

Theorem 3.1.2 of Cover & Thomas¹² (p. 59) states that:

$$\Pr\left(A_\varepsilon^{(N)}\right) \geq 1 - \varepsilon$$

and

$$(1 - \varepsilon)e^{N(S-\varepsilon)} \leq \left|A_\varepsilon^{(N)}\right| \leq e^{N(S+\varepsilon)}$$

where \Pr is the probability distribution over N -tuples, for arbitrarily small ε provided that N is sufficiently large. Colloquially, this means that, for N large, almost all samples of size N are part of a typical set whose size is $\left|A_\varepsilon^{(N)}\right| \approx \exp(S)^N$.

Instead of considering the most typical sets, Theorem 3.3.1 (p. 63) considers the most probable set $B_\delta^{(N)}$, which comprises the most probable N -tuples such that $\Pr\left(B_\delta^{(N)}\right) \geq 1 - \delta$, with δ arbitrarily small. The theorem states that

$$\left|B_\delta^{(N)}\right| > e^{N(S-\eta)}$$

where both δ and η may be arbitrarily small provided that N is sufficiently large. Colloquially, this means that any arbitrarily large fraction of the most frequent N -tuples comprises at least $\exp(S)^N$ elements, for N large enough.

Taken together, these two theorems consistently show that the probability of N -tuples becomes increasingly uniform and behaves like $(1/\Omega)^N$ with $\Omega = \exp(S)$ as N increases (formula 3.27, p. 63, Cover and Thomas). In other words, N -tuple probabilities behave as if one were sampling N times from a uniform distribution of size $\exp(S)$. A possible way to understand this result is that, when probabilities of single draws are uneven, in long series, it becomes exponentially less likely to draw an element outside of a certain set of size Ω as compared to resampling an element that has already been seen because it is more probable.

The same notions admit an interpretation in thermodynamics, which we briefly present as it may help readers familiar with statistical physics, following Kardar¹³ (pp. 110-113). Consider a system described by the Boltzmann probability distribution:

$$P(x) \sim \exp\{-H(x)\}$$

The microstates are no longer equiprobable; their probability is determined by the energy function $H(x)$ (in our case, Profile, SB, DCA, and DCA-SB correspond to this case). Note that all the system microstates with the same energy have the same assigned probability. The energy

function is extensive, meaning that it scales linearly with the size of the system L (the sequence length for an RNA system), i.e.,

$$H(x) = L \cdot e(x)$$

With $e(x)$ being the energy density. In the thermodynamic limit (i.e., $L \rightarrow \infty$) there exists an energy density value e_{avg} around which the state probability distribution becomes sharply peaked. Thus, drawing from the probability distribution yields a state with energy density close to e_{avg} with probability tending to 1. The distribution entropy S in this case corresponds to

$$S \approx \ln \Omega(e_{avg})$$

Since a finite sample will never exhibit extensive energy differences (or finite differences in energy density), the sampled states, with probability close to 1, will belong to the set of $\Omega(e_{avg}) = \exp(S)$ states around e_{avg} .

Below, for some models, we could compute the support size exactly from combinatorics, in other cases we computed approximate estimations, at any given mutational distance K . However, in some cases, such as the VAE model, there is no known strategy to compute the effective support size.”

[...] Although it is not known how to compute Ω in general, we devised a semi-analytical method exploiting the structure of DCA.”). While the results sound impressive (“The success rate at $L_{max}=60$ mutations was determined to be larger than 1% with $p < 126 \cdot 4 \cdot 10^{-5}$ confidence.”) I don't understand the procedure that was described in this paragraph.

We have expanded the corresponding sentence:

“The full process consists of computing the effective number of sequences $\Omega(L)$ that can be generated by the model (e.g., DCA) at a mutational distance L , then multiplying it by the active fraction experimentally measured at L . The maximum of the number at the particular distance L constitutes a lower bound for the whole neutral-set size Ω . In practice, this maximum was determined as the largest L such that we could measure the active fraction to be significantly larger than zero. Although it is not generally known how to compute Ω for probabilistic or generative models, we devised a semi-analytical method exploiting the structure of DCA (Supplementary Table S2, Supplementary Information ‘Support size computations’)³², which we applied at each L to obtain $\Omega(L)$.”

Note that the semi-analytical method is reported in the Supplementary Information, section ‘Support size computations’, subsection ‘Energy-based models’. It is detailed as well in the reference we cite: Calvanese et al. Nucleic Acid Research 2024. Computation of significance levels is described at the end of the last section of the Methods (‘L50, Lmax, and significance levels’).

The paragraph starting in line 128 seems to describe that the most important factor for activity is the closeness to the sequence of the Azoarcus ribozyme (second sentence). The next sentence

("To quantify the variety of sequences generated relative to the 132 natural diversity, ") took me a while to understand - please make it simpler.

We reworked the beginning of this paragraph, which we hope is now easier to understand:

“We found that the chimeras (representative of natural diversity) clustered near the Azoarcus ribozyme in these projections (Fig. 2c), while the DCA-generated sequences bridge the sequence space between the Azoarcus ribozyme and natural chimeras, from the perspective of the first two principal components. We note that PC1 strongly correlates with the distance to the Azoarcus ribozyme (Pearson $\rho=0.88$, $p<10^{-5}$). To further compare the variety of sequences generated by each model, we computed two different mutational distance distributions. First, we computed the distance from each DCA sequence to the closest chimera, including Azo. This can be interpreted as a measurement of how different the samples are from any natural sequences (Fig. 3a, top). Second, we computed the distances between all pairs of sequences within each model (Fig. 3a, bottom).”

At the end of this paragraph, please clarify also which results are directly supported by wet-lab analysis, and which are computational results.

We have edited the paragraph to clarify this:

“Experimentally active DCA sequences were found to be on average 25 mutations away from each other (Fig. 3a, bottom), on average 18 mutations away from any chimera, and up to 55 mutations away from any chimera (Fig. 3a, top, Supplementary Table 3). These distances were comparable to those found between all chimeras (active and non-active), which had 32 mutations between them on average and up to 59 mutations away from each other (Fig. 3a, Supplementary Table 3). It thus appears that DCA achieves a form of interpolation, by generating a diversity of experimentally active sequences similar to the diversity of natural Group I introns (Fig. 3bc), as well as a form of extrapolation, by generating active sequences as far away from any chimeras as the latter are from each other. Extrapolation is further confirmed by the largest distance between active DCA designs and chimeras being 99 mutations (Supplementary Table 3), well beyond the largest distance between any two chimeras.”

3 - Lines 219-233: The results are over-interpreted with respect to origins of life (for example line 233: "...making evidence for a probable RNA origin of life within reach."). First, the self-reproducing ligase ribozymes and Azoarcus ribozymes discussed in this study can only reproduce themselves from RNA fragments with very specific sequences. In a prebiotic environment it is hard to envision how such fragments could have existed, and in the absence of such fragments, no replication results. Second, self-replication does not equal life. Without open-ended evolution, the replicating system, unable to invent new catalytic activities, eventually dies out. Additionally, the results found with the Azoarcus ribozyme are non-quantitative, and the current success of the model is based on a few instances among the 24,000 tested RNA sequences.

We agree with Azoarcus only being a model and addressing one step during abiogenesis. To address similar comments of Reviewer 1, we have significantly expanded the discussion to comment on these points, clarifying or toning down some of the claims, but have also added results on the generality of diversification estimations to other RNAs. These modifications are presented in answers to points 2 and 5 of Reviewer 1.

To improve confidence in our result, we did an additional experimental with a pool of 1000 sequences instead of 150 at 60 mutations from the Azoarcus reference, using DCA. This result confirms the initial one with an improved p-value. It is presented in a new last section of the results together with a new Figure 5.

“We next thought to strengthen our estimation of the size of the neutral set obtained with the DCA model. As mentioned, the estimation is based on the support size curve combined with the number of active sequences found at different mutational distances (Fig. 2), and is sensitive to deviations at the largest mutational distances. To better establish our best estimation of Ω_{DCA} , we generated a DCA pool of 1000 sequences at 60 mutations and experimentally determined the active fraction. The results showed that 2,2% of the sequences were functional, consistent with our initial estimate. This result strengthens our initial sampling, with a strong level of significance, leading to a p-value $< 10^{-21}$ for the 60-mutation bin (Fig. 5a).”

Fig. 5. Robustness and generality. **a**, Designed DCA sequences at 60 mutations from the WT. The histogram shows the activity scores of 991 detected designs (out of 1000 designs) with 22 active sequences beyond the activity threshold (equal to -2.76, red dotted line), corresponding to 2% of the pool. **b**, Comparison of active fraction as a function of the number of mutations for the RUM and DCA-SB models, for different MgCl₂ concentrations: 60mM as our standard condition (solid lines) and a lower concentration of 5 mM (dashed lines). Dots are the average activity per bin of 5 mutations and bars represent the 25-75 percentile range.

minor points

abstract, second-last line: typo: it should be "... can be addressed ..."

Thank you, corrected.

line 40: it should be '.. a neutral network...' or 'the corresponding neutral network' but not 'the neutral network' because it has not yet been specified in the introduction what neutral network it refers to.

Upon the request of Reviewer 1, we now define the neutral network, or neutral set, earlier in the introduction.

line 78: Is it a typo that the acronym RDM is used for 'random uniform mutagenesis'? Please make clear why RDM is the right acronym, or use RUM instead.

RUM is a more logical choice, which we applied throughout the text and figures.

The manuscript contains several three-letter acronyms that are unnecessary and make reading hard (RDM, DCA, MSA, CHI, BPR, VAE). Please minimize the use of acronyms. For example, there is no reason to use MSA; this acronym is used only once more in addition to its original use and definition, and it is much easier for the reader if the term is spelled out twice. CHI also seems unnecessary. While 'Azo' is often used by researchers in the field, please avoid this lab jargon and instead use the full name 'Azoarcus ribozyme'.

To minimize the use of acronyms, we have replaced all along:

- MSA by Multiple Sequence Alignment
- GIs by Group I introns
- Azo by Azoarcus ribozyme.

However, we have chosen to keep model acronyms for consistency, including chimeras, which can be seen as a null model together with the random one. We agree that the use of acronyms can be heavy, but it seems customary in the field of generative models, notably to ensure a systematic comparison in the presentation of results in figures. We have added a new Table 1 that reports acronyms and their meaning.

Reviewer #1 (Remarks to the Author):

The authors have addressed almost all my points satisfactorily. However, I remain unconvinced of the necessity and usefulness of the probability calculation as it is currently presented in the discussion. I agree with the new (and well-justified) statements that this number should not be taken literally, since it should not be interpreted as a meaningful measure of abiogenic plausibility. But what conceptual purpose does it serve then? In fact, this kind of framing is prone to misinterpretation, especially in public or interdisciplinary discourse, where such probabilistic arguments have been (mis)used in the past, e.g. to support intelligent design narratives.

Despite the subsequent qualifications, the limitations appear almost as afterthoughts, logically subordinate to a statistical framework that itself rests on highly constrained and arguably artificial assumptions. In my view, by including this calculation, the authors risk reinforcing the very misconception they are trying to disarm: that the origin of life depends on the improbable spontaneous appearance of a highly complex, functional replicator.

I therefore strongly recommend reorienting the discussion to focus on the compelling empirical and modeling results regarding ribozyme diversity and either remove the probability estimate altogether and discussing things more qualitatively, or adding or combining it with probability estimates of much shorter replicators (which have been found experimentally, as the authors' themselves point out) by extrapolating their findings from the azoarcus-type ribozymes to simpler ribozymes. This would allow the manuscript's real contributions (namely, the generative modeling and exploration of neutral spaces using high-throughput methods) to occupy the conceptual center they deserve, while avoiding the risk of unintentionally perpetuating outdated or misleading views of abiogenesis.

We understand the concern of the reviewer. Conveying a simple metric such as the density of self-reproducers is a double-edged sword: while it is easy to grasp, it can lead to misinterpretation. As it is only a secondary and illustrative aspect of the manuscript, we removed the corresponding section from the Supplementary Information and any mention of it from the Discussion. The latter has been modified for broader access following the recommendations of Reviewer 1 and Reviewer 3. The changes can be seen in red in the updated manuscript. We have also nuanced the motivation in the introduction accordingly. We highlight below an extract of these modifications relevant to the reviewer's request (end of the first paragraph of the discussion):

“While some ribozymes relevant to the origin of life still need to be discovered, another important question is whether there exists a diversity of sequences that can carry out a given activity among all possible sequences (the sequence space), an ensemble called the *neutral network*, or *neutral set*¹². In particular, the larger the diversity of self-reproducing RNAs, the more likely are transitions between self-reproducing systems, thus enabling primordial modes of evolution¹³.”

Reviewer #2 (Remarks to the Author):

Unavailable for the second round.

Reviewer #3 (Remarks to the Author):

In this study, Lambert et al. tested different models for generating active ribozyme RNA sequences from a reference self-reproducing ribozyme. The active sequences were identified using a high-throughput assay coupled with deep sequencing. The combination of the evolutionary model DCA with a structure-based score (SB) showed the best performance in generating long-distance mutations that retained ribozyme activity. The neutral space was computed after correction using experimental data. This study is experimentally sound; however, beyond providing a lower-bound estimate for the number of active ribozymes (as better models might yield even higher numbers), the new insights this work contributes to the ribozyme or origin-of-life research community should be described more clearly.

First, we thank the reviewer for her or his assessment and suggestions. We reworked the introduction and discussion to better link the challenges and purpose in origin of life research, and convey how the current developments in ML RNA generative design could contribute to it, citing some recent methodological contributions in the field. The modified sections appear in red in the revised Introduction and Discussion.

Other detailed Comments:

1. The title “Discovering self-reproducing ribozymes...” is broad and could imply, for example, that the presented models generate ribozyme variants with improved catalytic activity or kinetics over the reference sequence, or that they create active ribozymes from partially fixed folding scaffolds. However, this work demonstrates that the models generate sequences with long-distance mutations that retain partial activity of the reference ribozyme from an MSA input. Other properties such as folding robustness or thermostability—important characteristics in prebiotic evolutionary processes—were not analyzed or discussed. In my opinion, the title should be revised to better reflect the actual content of the research, such as:

“Exploring the Activity Neutral Network of Self-Reproducing Ribozymes...”

We understand the possible confusion, we have changed the title to:

“Exploring the space of self-reproducing RNAs using generative models”

There is a trade-off between making the title specific yet understandable to a broad audience. We believe that ‘exploring’ as proposed by the reviewer is indeed less suggestive of novel structures or optimization than ‘discovering’. Regarding the concept of neutral network, it is a rather technical term aimed at experts. In fact, we were asked to not use it before defining it in the first round of reviews. Additionally, for some researchers, it is only defined in the context of RNA secondary structure, where it was introduced. So, we decided to not put it in the title, and modified the abstract so it is clear what we achieve:

“Estimating the plausibility of RNA self-reproduction is central to origin-of-life scenarios. However, this property has been shown in only a handful of catalytic RNAs. **Here, we compared models for their generative power in diversifying a reference ribozyme**, based on statistical

covariation and secondary structure prediction, and experimentally tested model predictions using high-throughput sequencing. “

2. The correlation between the gel shift assay–verified activity and the activity score or model-generated score (if available) should be presented and discussed.

We have analyzed the PAGE gels of the isolated variants of Figure S2 (formerly S3), and plotted the band intensity corresponding to reaction products. These results are now presented as a new panel of Fig. S3 (formerly S4), shown below.

Note that sequencing has a broader dynamical range and can detect products that do not yield visible PAGE bands, hence the categorization into dot colors corresponding to the PAGE panels of Figure S2. In summary, red crosses correspond to inactive mutants according to the gel, thus their X-value is 0, and consistently, their sequencing score is below the sequencing activity threshold (dotted line). Other colors (blue crosses, yellow stars, green dots) are mutants that display visible bands, but more or less visible products of step 1 and step 2. In all cases, the total product intensity normalized by the initial mutant concentration correlates well with the sequencing score (computed for active sequences only, with a Pearson correlation of 0.61, p-value = 0.008, N=17). We added a corresponding discussion at the end of the first paragraph of the ‘High-throughput sequencing assay’ section of the text:

Additionally, we verified *a posteriori* that cross-catalysis negligibly affected the activity score by assaying variants separately by denaturing gel electrophoresis (Supplementary Fig. 2) batches (Supplementary Figure 3), finding that band intensities significantly correlated with sequencing scores (Pearson correlation of 0.61, p-value = 0.008, N=17). Consistently, mutants that displayed no visible product on the gel had a sequencing score below the activity threshold (Supplementary Fig. 3). We furthermore tested that incubation in sub-pools yielded the same sequencing scores as within the total pool (Supplementary Fig. 3, Methods ‘Cross-catalysis tests’).

Regarding the correlation of the experimental score with the DCA score, the mutants we had chosen for independent measurements covered a range of measured activities but were (unfortunately) all picked among good DCA scores (meaning low DCA score) as we were focused on a posteriori cross-validation between experimental measurements. For the information of the reviewer, we have embedded those mutants in Supplementary Fig. 7b (formerly SI Fig. 8b):

Nevertheless, this Supplementary Figure 7b shows more broadly the correspondence between the DCA score and the sequencing score. We hope that the combination of Supplementary Fig. 3a and Supplementary Fig. 7b will satisfy the reviewer.

3. In Figures 2a–b and 5b, only the active fraction is shown, which omits the information on the activity levels of the variants. The distribution of activity scores, or the mean/median activity score (as shown in Fig. S1), should be included in the main text to provide a more comprehensive view of catalytic activity.

As requested, we have moved Figure S1 as panel a of Figure 2, a meaningful choice as this panel is called multiple times throughout the main text.

4. A discussion on measuring the activity of the designed sequences at different (especially higher) temperatures, or performing the high-throughput assay under varied temperature conditions, could be added. This might offer additional insights into the structure of the neutral set and potential evolutionary trajectories.

This is an interesting suggestion. We performed an additional experiment at 60°C, significantly above the reference temperature of 37°C, and not far from the practical limit for *Azoarcus* (see gels below, where *Azoarcus* GII half-life is ~30 minutes at 70°C).

Note that we did not have access to a NovaSeq at this moment as for other experiments, and we used a lower depth MiSeq instead, thus used a more stringent activity threshold (-1.5) to account for increased noise, as reported in the text. To obtain a consistent comparison, we re-did the standard DCA at 37°C and the same pool at 60°C. The scatter plot of activity score at 37°C versus 60°C is now presented in Fig. 5c. The active fraction as a function of the number of mutations is shown in 5d, displaying a highly similar trend but lower L metrics, as it might be expected.

c, Sequencing score at 37°C versus 60°C for the same DCA pool. Due to lower sequencing depth as compared to assays of Fig. 2, a more stringent activity threshold of -1.5 was used (see Methods). The Pearson correlation is 0.64 with p -value $< 10^{-5}$, $N=558$, comprising 484 active for both conditions (upper-right quadrant), 25 inactive for both (lower-left), 19 active at 60°C but not 37°C, 30 active at 37°C but not 60°C. **d**, Active fraction as a function of the number of mutations for the DCA model at 37°C (plain line) and 60°C (dashed line). Note that the fraction is computed over all designed sequences, including those that were not active enough to be detected by sequencing (thus counted as inactive).

A corresponding section has been added in the main text, as follows:

Regarding the effect of temperature, the *Azoarcus* ribozyme strongly degrades above 70°C (half-life shorter than 30 minutes). We thus tested the standard DCA model pool at 60 °C, below this upper limit, but well above the 37°C of the initial test. Sequencing scores correlated well between the 2 conditions ($r=0.64$, p -value $< 10^{-5}$, Fig. 5c). The region below the $x=y$ diagonal appears more populated than above it (Fig. 5c), confirming the expectation that a given sequence should have a better activity at 37°C than at 60°C. Qualitatively, a large majority of the mutants detected by

sequencing were found to be above the activity threshold for both temperatures (484 over 558 detected sequences, those below the threshold or non-detected being considered inactive). Consistently, the active fraction as a function of the number of mutations was slightly lower at 60°C than 37°C, yet both temperatures followed the same trend (Fig. 5d), leading to $L_{50}=15$ and $L_{max}=45$ at 60°C (compared to $L_{50}=20$ and $L_{max}=60$ at 37°C).

Finally, we have updated the download link with these sequences, for anyone who wish to further investigate the effect of temperature.